# When Attributes Disagree:
# Gradient Conflict in Image Aesthetic Assessment

**Ye Wang**[1]  **Maocai Dai**[1]  **Jiang Xie**[1]  **Xiuli Bi**[* 1]  **Fei Tao**[2]  **Xiao Li**[3]  **Hong Yu**[1]

## Abstract

Image Aesthetic Assessment (IAA) predicts an image's overall aesthetic score, yet aesthetic is influenced by multiple attributes whose relative importance varies with image content and usage scenarios. Under end-to-end training with only overall-score supervision, attribute signals are blended, which can cause gradient conflict across samples dominated by different attributes, resulting in gradient cancellation and persistent systematic bias. To address these issues, we propose AGREE (Attribute-guided Gradient Routing for Establishing Agreement), which learns attribute-specific subspaces and performs gradient routing based on sample-wise attribute sensitivity estimated via perturbation analysis. AGREE further reduces feature coupling across attributes with semantic anchors and improves robustness via error-aware reweighting. Experiments on AVA, LAPIS, AADB, TAD66K, and PARA show consistent improvements over diverse IAA baseline models, and AGREE is plug-and-play for existing end-to-end IAA methods without modifying their original architectures. To our knowledge, this work is among the early efforts in IAA to systematically study gradient conflict and provide an effective solution. The code is available at `https://dahat364.github.io/AGREE/`.

## 1. Introduction

Image Aesthetic Assessment (IAA) aims to predict an image's overall aesthetic score and has been widely used in photography and content recommendation (Deng et al.,

[1]Key Laboratory of Cyberspace Big Data Intelligent Security, Ministry of Education; School of Computer Science and Technology; Chongqing University of Posts and Telecommunications, Chongqing, China. [2]NewsBreak, CA, USA. [3]Hong Kong Polytechnic University, China; University of Oxford, Oxford, UK. Corresponding Author: Xiuli Bi <bixl@cqupt.edu.cn>.

*Proceedings of the $43^{rd}$ International Conference on Machine Learning*, Seoul, South Korea. PMLR 306, 2026. Copyright 2026 by the author(s).

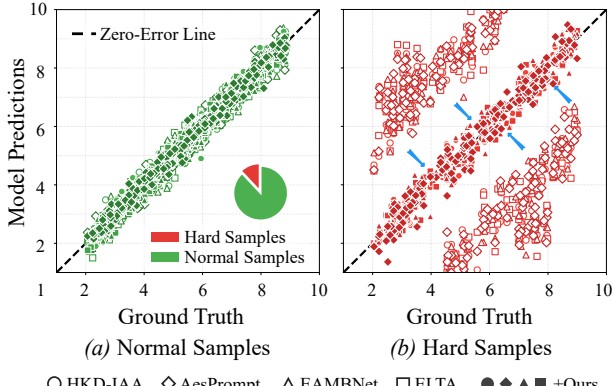

*Figure 1.* Baseline models (**hollow** markers) exhibit consistent error patterns on a subset of hard samples, which are substantially reduced after applying our method (+Ours, **filled** markers). The pie chart shows the proportion of normal (green) vs hard (red) samples. Normal samples refer to images that are consistently well predicted across multiple IAA models. Hard samples are defined as images that incur consistently large prediction errors across multiple IAA models.

2017; Zeng et al., 2019; Murray et al., 2012). Meanwhile, aesthetic quality is inherently determined by multiple attributes, such as color, sharpness, and composition, whose relative importance varies across usage scenarios (Jin et al., 2019; Pan et al., 2019; Yang et al., 2022). Therefore, the contribution of individual attributes to the final aesthetic score is not directly observable under overall-score supervision.

By inspecting error distributions of several mainstream IAA methods, Figure 1(a) shows that different IAA models achieve competitive overall performance on normal samples. However, Figure 1(b) reveals a contrasting phenomenon: all models perform poorly on hard samples, exhibiting consistent, cross-model systematic error patterns. Furthermore, we identify that certain training interventions can move these clusters toward the zero-error line. This raises the question of why such consistent biases emerge under standard end-to-end IAA training.

We consider this phenomenon as an optimization-level issue under overall-score supervision. Most IAA methods learn all images with a single parameter set $\theta$ (i.e., a shared parameter space) (Ma et al., 2017; Talebi & Milanfar, 2018).

Yet different image subsets rely on different attributes (Pan et al., 2019; Yang et al., 2022). A single shared update may benefit one subset while contributing little to another. Over the training, updates tend to be influenced more by subsets associated with dominant attributes, leaving some samples persistently hard to optimize. Empirically, such persistently hard samples often correspond to less dominant attributes in the training data. We further analyze this phenomenon in Section 5.4.

A natural idea is to strengthen each sample's most relevant representations, reducing cross-attribute interference and making updates more consistent. However, under overall-score supervision, sample-wise sensitivity to different attributes is not directly observable from the overall score alone (Kong et al., 2016; Huang et al., 2024a). Thus, overall-score supervision requires: (i) mitigating interference across attributes, and (ii) estimating each sample's relative sensitivity to different attributes without explicit labels.

For this challenge, we propose AGREE (Attribute-guided Gradient Routing for Establishing Agreement), a plug-and-play framework. AGREE learns attribute-related representations and performs sensitivity-guided update allocation at the sample level. To mitigate cross-attribute interference, AGREE introduces inductive bias via attribute-specific image transforms and separate adapters, and further reduces feature coupling using semantic anchors. To estimate sample-wise attribute sensitivity without labels, AGREE leverages perturbation-based analysis and uses the resulting signals to guide feature fusion and update allocation. In addition, AGREE incorporates an error-aware reweighting strategy to emphasize persistently hard samples during training.

Our contributions are summarized below:

- We provide an optimization-oriented analysis of image aesthetic assessment under overall-score supervision, revealing that shared-parameter training can lead to persistent, attribute-related biases across different models.
- We propose AGREE, a plug-and-play framework that adapts update allocation to sample-specific attribute relevance, and can be seamlessly integrated into existing IAA models without additional supervision.
- Through comprehensive experiments on diverse datasets, we show that the proposed AGREE consistently improves performance and substantially reduces errors on hard-to-optimize samples.

## 2. Related Work

### 2.1. Mainstream end-to-end IAA methods

Mainstream end-to-end IAA methods extract visual representations with CNNs or ViTs and predict an overall aes-

thetic score via regression, ranking, or distribution modeling (Daryanavard Chounchenani et al., 2025; He et al., 2022). Despite scalar supervision, aesthetic quality depends on multiple attributes (Celona et al., 2022; Zhu et al., 2020), and the dominant attribute can vary substantially across images and scenarios (Yun & Choo, 2025; Behrad et al., 2025). Consequently, under overall-score supervision with shared parameters, attribute-related cues are implicitly entangled in a single objective (Datta et al., 2006; Malu et al., 2017).

Several studies report uneven optimization across samples under shared training (Daryanavard Chounchenani et al., 2025; Lu et al., 2014). Updates may be driven primarily by dominant attribute cues (Sheng et al., 2020; Soydaner & Wagemans, 2024), while samples dominated by less prevalent attributes are harder to optimize and exhibit systematic bias (Xu et al., 2025; Huang et al., 2024b). Related multimodal robustness studies show that shortcut correlations can cause systematic failures under distribution shifts (Ma et al., 2024), while adversarial studies reveal fragile cues in black-box IAA models (He et al., 2025b). Together, these findings suggest that strong overall performance may mask sample-specific reliability gaps caused by sample-dependent attribute reliance.

### 2.2. Attribute-Aware and Multi-Task Methods

Attribute-aware and multi-task methods treat aesthetic attributes as auxiliary targets, typically via multi-head architectures with additional supervision (Jin et al., 2019; Pan et al., 2019). By disentangling attribute-specific signals, they encourage capacity allocation across diverse attributes (Yan et al., 2022; Soydaner & Wagemans, 2024) and may mitigate training interference. However, most IAA benchmarks provide only overall-score scalars (He et al., 2022), so attribute labels, auxiliary datasets, or modified protocols are required to apply these methods (Celona et al., 2022; Sheng et al., 2025a). This supervision mismatch limits direct applicability and comparability in the overall-score-only setting.

Recent works have explored learning attribute relevance or sensitivity from weak or implicit supervision (Huang et al., 2024a; Fontana et al., 2024), aiming to recover attribute-related structure without explicit labels (He et al., 2025a; Yun & Choo, 2025). These efforts leverage the attribute structure under overall-score supervision while avoiding additional annotations.

### 2.3. Optimization Coordination in Multi-Task Learning

In multi-task learning, update interference across tasks is a well-studied issue (Désidéri, 2012). It is commonly described as gradient conflict, and various coordination strategies have been proposed, including task reweighting (Chen et al., 2018), gradient projection (Yu et al., 2020), and multi-objective optimization (Liu et al., 2021). These methods

typically assume the availability of explicit task identities and per-task losses (Hu et al., 2022), and coordinate learning through task-level gradients (Chen & Er, 2025).

Beyond standard multi-task learning, recent vision-language studies have explored conflict identification and neutralization in compositional retrieval (Lin et al., 2025; Tian et al., 2025), showing that semantic conflicts can affect representation learning and retrieval behavior. However, they mainly target explicit retrieval conflicts, whereas overall-score-only IAA involves implicit attribute-induced optimization conflicts. Moreover, global coordination objectives are less suited to sample-level attribute heterogeneity, motivating formulations that infer attribute relevance and mitigate interference without explicit task definitions.

# 3. Theoretical Analysis

**Scope.** We provide an explanation-oriented analysis of standard overall-score IAA training. Without introducing new objectives, we characterize when shared-parameter optimization can exhibit interference across attribute-dominant subsets and use the insights to motivate Section 4. Our analysis is intended as intuition under mild conditions rather than a general guarantee.

## 3.1. Problem Setup

### 3.1.1. UPDATE INTERFERENCE AND GRADIENT CONFLICT (ANALYSIS TERM)

IAA learns a mapping $f_\theta : \mathbb{R}^{H \times W \times 3} \to \mathbb{R}$ from an image $x$ to an aesthetic score $\hat{y}$. Given $\mathcal{D} = \{(x_i, y_i)\}_{i=1}^N$ with normalized labels $y_i \in [0, 1]$, conventional training minimizes

$$\theta^* = \arg\min_\theta \mathcal{L}(\theta) = \arg\min_\theta \frac{1}{N} \sum_{i=1}^N \ell(f_\theta(x_i), y_i), \quad (1)$$

where $\ell$ is typically MSE.

Aesthetic quality depends on multiple attributes. In principle, the analysis applies to any predefined attribute partition; in this work, we instantiate it with five widely used ones—**Brightness**, **Contrast**, **Blur**, **Hue**, and **Saturation** (Datta et al., 2006; Ke et al., 2006; Hasler & Suesstrunk, 2003; Kong et al., 2016)—and set $K = 5$. Let $s_k(x) \geq 0$ denote the sensitivity of $x$ to attribute $k$, and define the dominant attribute

$$k^*(x) = \arg\max_k s_k(x). \quad (2)$$

This induces a partition

$$\mathcal{D} = \bigcup_{m=1}^M \mathcal{D}_m, \qquad \mathcal{D}_m = \{x \in \mathcal{D} : k^*(x) = m\}, \quad (3)$$

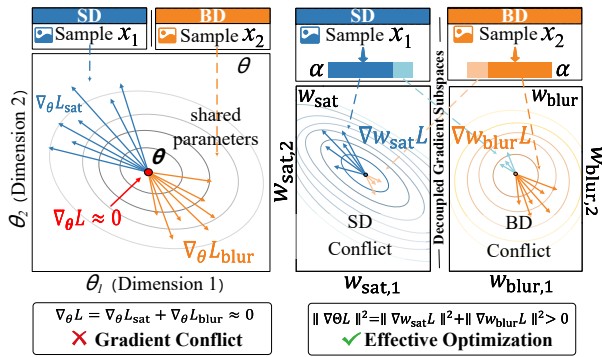

*Figure 2.* Illustration of gradient conflict and decoupled optimization between SD and BD subsets. Left: opposing gradients on shared parameters cause cancellation. Right: decoupled subspaces reduce interference and enable effective updates. **SD** denotes Saturation Dominant, and **BD** denotes Blur Dominant.

and the subset loss is defined as

$$\mathcal{L}_m(\theta) = \frac{1}{|\mathcal{D}_m|} \sum_{(x,y) \in \mathcal{D}_m} \ell(f_\theta(x), y). \quad (4)$$

The overall gradient is a weighted vector sum of subset gradients:

$$\nabla_\theta \mathcal{L}(\theta) = \sum_{m=1}^M p_m \nabla_\theta \mathcal{L}_m(\theta), \quad (5)$$

where $p_m = |\mathcal{D}_m|/N$. When subset gradients point in opposing directions, they can partially cancel.

**Definition 3.1. Gradient conflict (full space).** We define gradient conflict between two subsets $\mathcal{D}_i, \mathcal{D}_j$ at $\theta$ to occur if $\langle \nabla_\theta \mathcal{L}_i(\theta), \nabla_\theta \mathcal{L}_j(\theta) \rangle < 0$, as illustrated in Figure 2(left).

**Definition 3.2. Coupled parameters across attributes.** Let the dominant attributes of $\mathcal{D}_i$ and $\mathcal{D}_j$ be $k$ and $k' \neq k$. Parameters are *coupled across* $(k, k')$ if there exists a coordinate $\theta_c$ that affects subset losses, i.e., $\nabla_{\theta_c} \mathcal{L}_i(\theta) \neq 0$ and $\nabla_{\theta_c} \mathcal{L}_j(\theta) \neq 0$ in a neighborhood of interest.

**Assumption 3.3. Illustrative opposite preference on a coupled coordinate.** Assume there exists a coupled coordinate $\theta_c$ (as in Definition 3.2) such that

$$\frac{\partial \mathcal{L}_i(\theta)}{\partial \theta_c} < 0 \quad \text{and} \quad \frac{\partial \mathcal{L}_j(\theta)}{\partial \theta_c} > 0. \quad (6)$$

Under Assumption 3.3, the two subsets conflict on the coupled coordinate $\theta_c$ since $(\partial \mathcal{L}_i / \partial \theta_c)(\partial \mathcal{L}_j / \partial \theta_c) < 0$. A discussion of when coordinate-level conflict implies full-space conflict is provided in Appendix C.1.

### 3.1.2. OPTIMIZATION STAGNATION UNDER SHARED PARAMETERS

We illustrate a failure mode where interference across subsets makes the overall gradient small even when subset gra-

dients remain non-negligible, as shown in Figure 2(right).

**Definition 3.4. Oracle decoupled baseline.**

$$\mathcal{L}_{\text{ideal}} = \sum_{m=1}^{M} p_m \cdot \min_{\theta_m} \mathcal{L}_m(\theta_m), \qquad (7)$$

We next give an illustrative proposition showing how cancellation can lead to (near-)stationarity.

**Proposition 3.5.** *Cancellation and approximate stationarity.* *Let $\mathcal{L}(\theta) = \sum_{m=1}^{M} p_m \mathcal{L}_m(\theta)$. Suppose for two subsets $i, j$ there exists $c > 0$ such that $\left\| c \nabla_\theta \mathcal{L}_i(\theta) + \nabla_\theta \mathcal{L}_j(\theta) \right\| \leq \delta$ and $\left\| \sum_{m \neq i,j} p_m \nabla_\theta \mathcal{L}_m(\theta) \right\| \leq \varepsilon$. Then*

$$\|\nabla_\theta \mathcal{L}(\theta)\| \leq |p_i - cp_j| \, \|\nabla_\theta \mathcal{L}_i(\theta)\| + p_j \delta + \varepsilon. \quad (8)$$

*When $p_i \approx cp_j$ and $\delta, \varepsilon$ are small, $\theta$ can be (near-)stationary even if $\|\nabla_\theta \mathcal{L}_i(\theta)\|$ and $\|\nabla_\theta \mathcal{L}_j(\theta)\|$ are not small.*

*Proof.* Rewrite

$$R(\theta) := \sum_{m \neq i,j} p_m \nabla_\theta \mathcal{L}_m(\theta).$$

$$\nabla_\theta \mathcal{L} = (p_i - cp_j)\nabla_\theta \mathcal{L}_i + p_j \big( c\nabla_\theta \mathcal{L}_i + \nabla_\theta \mathcal{L}_j \big) + R(\theta).$$

Then apply the triangle inequality and the two assumed bounds to obtain (8). **See Appendix C.2 for details.** □

**Proposition 3.6.** *Oracle lower bound and unattainability.* *For any shared parameter $\theta$, we have $\mathcal{L}(\theta) \geq \mathcal{L}_{\text{ideal}}$. Moreover, if $\mathcal{L}_{\text{ideal}}$ is unattainable by any shared $\theta$, then $\mathcal{L}(\theta) > \mathcal{L}_{\text{ideal}}$ for all shared $\theta$; in particular, any stationary point $\theta^*$ of $\mathcal{L}$ satisfies $\mathcal{L}(\theta^*) > \mathcal{L}_{\text{ideal}}$.*

*Proof.* Summing $p_m \mathcal{L}_m(\theta) \geq p_m \min_{\theta_m} \mathcal{L}_m(\theta_m)$ over $m$ yields $\mathcal{L}(\theta) \geq \mathcal{L}_{\text{ideal}}$. If $\mathcal{L}_{\text{ideal}}$ is unattainable by any shared $\theta$, then $\mathcal{L}(\theta) > \mathcal{L}_{\text{ideal}}$ for all shared $\theta$, and therefore for any stationary point under shared parameters. □

### 3.2. Decoupled Blocks and Sample-wise Allocation

The analysis suggests two sources of interference: parameter coupling across attributes and sample-dependent attribute reliance under a single objective. A natural mitigation is to decouple attribute-related parameter blocks and allocate updates per sample based on attribute reliance; AGREE (Section 4) instantiates this design.

We write a generic prediction form as

$$\hat{y} = g_\Theta(x) = \text{Pred}(\text{Fuse}(h_0(x; w_0), z(x))), \quad (9)$$

where

$$z(x) = \sum_{k=1}^{K} \alpha_k(x) \, h_k(x; w_k), \qquad (10)$$

with $\Theta = \{w_0, w_1, \ldots, w_K\}$, $\sum_k \alpha_k(x) = 1$, and loss

$$\mathcal{L}(\Theta) = \sum_{i=1}^{N} \omega_i \cdot \ell(g_\Theta(x_i), y_i), \qquad (11)$$

where $\omega_i$ is an error-aware reweight.

**Intuition.** If $\alpha$ is sparse (with a few dominant weights) and different blocks are only weakly correlated, then cross-block interference is expected to be limited in practice.

**Theorem 3.7.** *Approximate orthogonality.* *For $i \neq j$, the gradient updates to blocks $w_i$ and $w_j$ are approximately decoupled:*

$$\langle \nabla_{w_i} \mathcal{L}, \, \nabla_{w_j} \mathcal{L} \rangle \approx 0. \qquad (12)$$

*Proof.* Let $r_n = \hat{y}_n - y_n$. By chain rule,

$$\nabla_{w_k} \mathcal{L} = \sum_n \omega_n \cdot 2 r_n \cdot \alpha_k(x_n) \cdot \frac{\partial h_k}{\partial w_k}. \qquad (13)$$

For $i \neq j$, embedding both gradients into the concatenated parameter space and assuming cross-sample Jacobian terms are negligible in high dimensions,

$$\langle \nabla_{w_i} \mathcal{L}, \, \nabla_{w_j} \mathcal{L} \rangle \approx \sum_n \omega_n^2 r_n^2 \, \alpha_i(x_n) \alpha_j(x_n) \left\langle \frac{\partial h_i}{\partial w_i}, \frac{\partial h_j}{\partial w_j} \right\rangle. \qquad (14)$$

When $\alpha_i(x_n)\alpha_j(x_n)$ is small for most samples and the Jacobians of different blocks are weakly aligned, the sum is small, yielding (12). **See Appendix C.3 for details.** □

**Corollary 3.8.** *Reduction of gradient conflict (approximately).* *For subsets $\mathcal{D}_i, \mathcal{D}_j$ dominated by attributes $i$ and $j$, and any $w_k$,*

$$\langle \nabla_{w_k} \mathcal{L}_{\mathcal{D}_i}, \, \nabla_{w_k} \mathcal{L}_{\mathcal{D}_j} \rangle \approx 0. \qquad (15)$$

*Proof.* If $k = i$ (resp. $k = j$), then $\alpha_i(x) \approx 0$ on $\mathcal{D}_j$ (resp. $\alpha_j(x) \approx 0$ on $\mathcal{D}_i$), so the cross-gradient on $w_k$ is $\approx 0$; if $k \neq i, j$, both subsets have negligible gradient on $w_k$. Summing over $k$, the full inner product $\langle \nabla_\Theta \mathcal{L}_{\mathcal{D}_i}, \nabla_\Theta \mathcal{L}_{\mathcal{D}_j} \rangle \approx 0$, so gradient conflict (Definition 3.1) is mitigated. □

Unlike the vector sum in (5) that can cancel, the squared norm now decomposes as a scalar sum.

**Corollary 3.9.** *Effective gradient updates (blockwise).* *With concatenated parameters $\Theta = \{w_0, w_1, \ldots, w_K\}$,*

$$\|\nabla_\Theta \mathcal{L}\|^2 = \sum_{k=0}^{K} \|\nabla_{w_k} \mathcal{L}\|^2. \qquad (16)$$

*Hence cross-block cancellation is structurally limited, and $\|\nabla_\Theta \mathcal{L}\| \approx 0$ typically requires all blocks to be (approximately) stationary.*

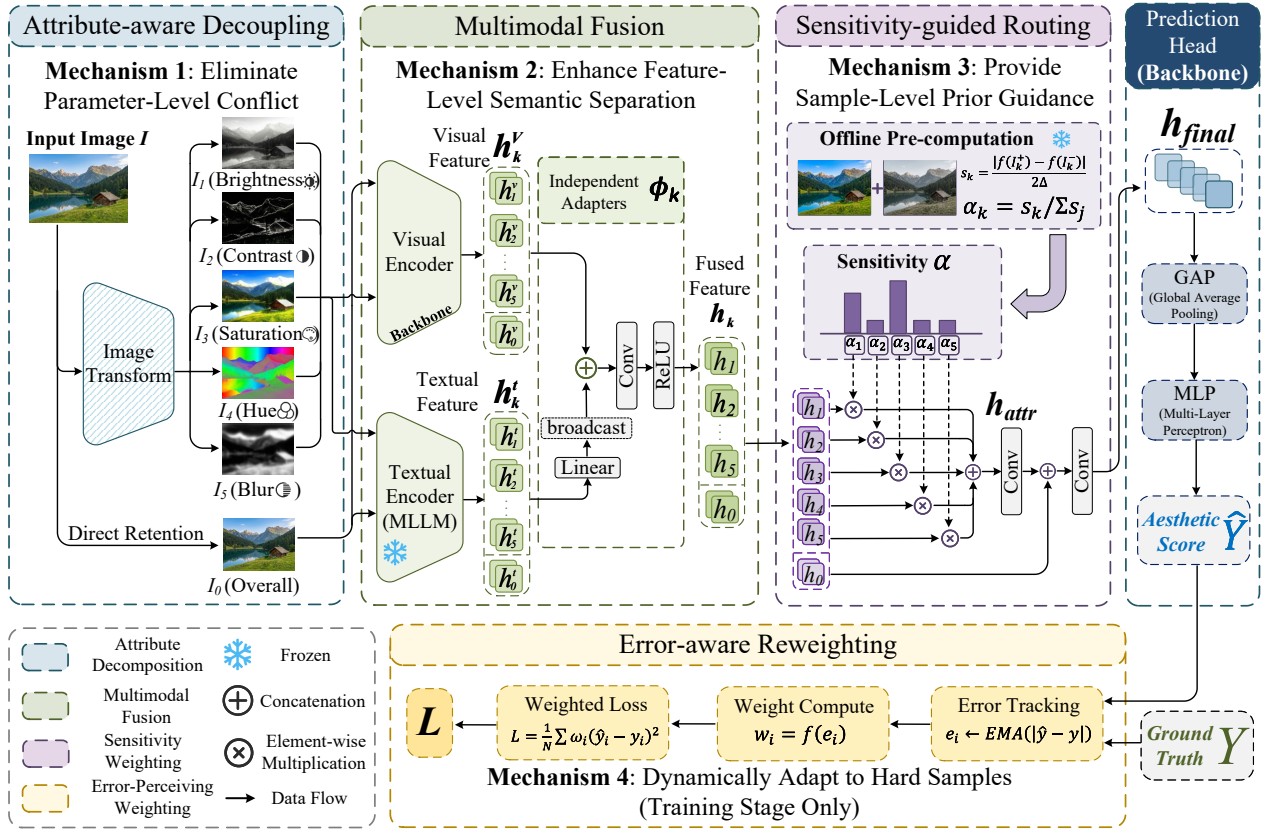

*Figure 3.* Overview of AGREE, a plug-and-play framework consisting of four mechanisms: (1) Attribute-aware Decoupling reduces parameter-level conflict via attribute-specific branches; (2) Multimodal Fusion injects frozen text anchors for semantic separation; (3) Sensitivity Weighting uses offline per-sample sensitivities to guide fusion; and (4) Error-aware Reweighting upweights hard samples using EMA-tracked errors during training.

*Proof.* Different $w_k$ occupy disjoint coordinate blocks in the concatenated space, so (16) follows directly from the Euclidean norm definition. **See Appendix C.4 for details.**

$\square$

# 4. Method

Based on the theoretical analysis in Section 3, gradient conflict mainly arises from parameter coupling and sample-dependent attribute reliance. As shown in Figure 3, AGREE mitigates such conflict via four mechanisms: (i) attribute decoupling (parameter level), (ii) multimodal fusion with text anchors (feature level), (iii) sensitivity-based weighting (sample level), and (iv) error-aware reweighting (loss level).

## 4.1. Attribute-aware Decoupling

To reduce parameter coupling, we decompose an image into attribute dimensions and model them with separate parameter blocks. Given $I \in \mathbb{R}^{H \times W \times 3}$, we construct attribute-augmented views

$$I_k = \mathcal{T}_k(I), \quad k = 1, \ldots, 5, \quad (17)$$

where $\mathcal{T}_k$ applies Brightness, Contrast, Blur, Hue, or Saturation. We then extract features with the baseline's encoder $\mathcal{E}$ and feed each view into an independent adapter:

$$h_k^v = \phi_k\big(\mathcal{E}(I_k)\big), \quad k = 0, \ldots, 5, \quad (18)$$

with $I_0 = I$. Each adapter $\phi_k$ has its own parameters $w_k$, yielding decoupled blocks $\Theta = \{w_0, w_1, \ldots, w_5\}$, where $h_0^v$ is overall and $\{h_k^v\}_{k=1}^5$ are attribute-specific.

## 4.2. Multimodal Fusion

To enhance semantic separation across attributes, we introduce text features as fixed semantic anchors. For each attribute $k$, we design a prompt $P_k$ describing its ideal aesthetic property, and extract the hidden state from the penultimate decoder layer of LLaVA:

$$h_k^t = \text{LLaVA}_{\text{hidden}}(P_k, I_k) \in \mathbb{R}^{D_t}, \quad (19)$$

where $D_t = 4096$. All $h_k^t$ are pre-extracted once and frozen. Importantly, LLaVA is used only in this offline anchor-extraction step and is not involved in AGREE training or inference. During training and inference, the text branch

only performs lightweight projection and fusion on these fixed vectors. We also include $P_0$ for the overall branch to provide a global semantic anchor. For an investigation of the effectiveness of the feature $h_0$, please refer to **Appendix B.1**.

Fusion proceeds as: (1) project text to channel dimension $\tilde{h}_k^t = W_t h_k^t + b_t \in \mathbb{R}^C$; (2) expand to spatial map $H_k^t = \text{Expand}(\tilde{h}_k^t) \in \mathbb{R}^{C \times H' \times W'}$; (3) concatenate and fuse with a $1 \times 1$ convolution:

$$h_k = \text{ReLU}\big(\text{Conv}_{1\times1}^{2C \to C}([h_k^v; H_k^t])\big) \in \mathbb{R}^{C \times H' \times W'}, \tag{20}$$

where $[\,;\,]$ denotes channel-wise concatenation.

### 4.3. Sensitivity-guided Routing

We next compute a model-dependent prediction-sensitivity signal for sample-wise routing. Following Section 3, we estimate how the baseline predictor $f_{\theta^*}$ responds to paired perturbations along each attribute:

$$s_k(I) = \frac{\left|f_{\theta^*}(I^{+k}) - f_{\theta^*}(I^{-k})\right|}{2\Delta}, \tag{21}$$

where $I^{+k} = \mathcal{T}_k^{+\Delta}(I)$ and $I^{-k} = \mathcal{T}_k^{-\Delta}(I)$. Importantly, $s_k(I)$ is not intended to represent ground-truth or causal aesthetic importance. It only measures the prediction sensitivity of the current baseline to attribute-$k$ perturbations and serves as an operational prior for routing attribute-specific features under overall-score-only supervision. We normalize these sensitivities into weights

$$\alpha_k(I) = \frac{s_k(I)}{\sum_{j=1}^{5} s_j(I)}, \tag{22}$$

with $\sum_{k=1}^{5} \alpha_k(I) = 1$. All $\alpha_k(I)$ are computed offline and fixed.

We fuse attribute features using $\alpha_k$, then combine them with the overall feature:

$$F_{\text{attr}} = \text{Conv}_{1\times1}^{5C \to C}\Big(\text{Concat}(\alpha_1 h_1, \alpha_2 h_2, \ldots, \alpha_5 h_5)\Big), \tag{23}$$

$$F_{\text{final}} = \text{Conv}_{1\times1}^{2C \to C}\Big(\text{Concat}(h_0, F_{\text{attr}})\Big), \tag{24}$$

where $h_0$ is kept unweighted to preserve global aesthetics. The final prediction is

$$\hat{y} = \text{MLP}(\text{GAP}(F_{\text{final}})), \tag{25}$$

with GAP as global average pooling.

### 4.4. Error-Aware Reweighting

Finally, we emphasize hard samples by dynamically reweighting the MSE loss using historical prediction

errors(EA-MSE). We track an EMA error per sample:

$$e_i^{(t)} = \gamma e_i^{(t-1)} + (1 - \gamma)\left|y_i - \hat{y}_i^{(t)}\right|, \tag{26}$$

with $\gamma = 0.9$. We then assign an error-aware reweight

$$\omega_i = 1 + \beta \cdot \tanh\left(\max\left(0, \frac{e_i - \mu_g}{\tau \sigma_g}\right)\right), \tag{27}$$

where $\mu_g, \sigma_g$ are the global mean/std of errors, $\beta = 1.0$, and $\tau = 0.1$, yielding $\omega_i \in [1, 1 + \beta]$. Using batch-normalized weights $\tilde{\omega}_i = \omega_i / \left(\frac{1}{B} \sum_j \omega_j\right)$, where $B$ is the batch size. Then, we minimize the loss as

$$\mathcal{L} = \frac{1}{N} \sum_{i=1}^{N} \tilde{\omega}_i (\hat{y}_i - y_i)^2, \tag{28}$$

## 5. Experiments

### 5.1. Experimental Setup

**Datasets and Metrics.** We evaluate on five IAA datasets: AVA (Murray et al., 2012), AADB (Kong et al., 2016), TAD66K (He et al., 2022), PARA (Yang et al., 2022), and LAPIS (Maerten et al., 2025). We report PLCC/SRCC, ACC, and MAE/RMSE. Preprocessing, splits, and metric definitions are in **Appendix A.1**.

**Baselines.** We compare with six representative IAA baselines: TANet (He et al., 2022), EAT (He et al., 2023), ELTA (Liu et al., 2024), EAMB-Net (Chen et al., 2024), HKD-IAA (Chen et al., 2025), and AesPrompt (Sheng et al., 2025b). We retrain all baselines under a unified protocol; "+Ours" only inserts AGREE while keeping the training budget and other hyperparameters unchanged. Details are in **Appendix A.2**.

**Implementation and Overhead.** All experiments use PyTorch on two RTX 4090 GPUs. AGREE adds $\sim$7.6M parameters (2.5–30% across backbones) with $< 10\%$ per-epoch training-time overhead; inference adds lightweight fusion, with one-time, label-free per-image sensitivity preprocessing offline. Details are in **Appendix A.3**.

### 5.2. Main Results

Table 1 summarizes results on five IAA benchmarks. AGREE achieves state-of-the-art SRCC/PLCC on all datasets when integrated with EAT or HKD-IAA. Relative improvements over the best baseline range from 1.6% to 4.8%, with notable gains on AVA and LAPIS, demonstrating strong generalization across general, topic-oriented, and personalized aesthetics.

*Table 1.* Main results (SRCC/PLCC) on AVA, LAPIS, PARA, AADB, and TAD66K. AGREE plugs into EAT and HKD-IAA, achieving the best results across all metrics. **Bold** indicates best results, underline indicates second-best.

| Method | AVA | | LAPIS | | PARA | | AADB | | TAD66K | |
|---|---|---|---|---|---|---|---|---|---|---|
| | SRCC | PLCC | SRCC | PLCC | SRCC | PLCC | SRCC | PLCC | SRCC | PLCC |
| TANet(22'IJCAI) | .684 | .675 | .694 | .706 | .815 | .853 | .564 | .575 | .425 | .457 |
| EAT(23'MM) | .752 | .755 | .802 | .809 | .891 | .924 | .601 | .611 | .476 | .503 |
| ELTA(24'ICML) | .698 | .704 | .686 | .696 | .826 | .851 | .592 | .583 | .413 | .430 |
| EAMBNet(24'TIM) | .702 | .707 | .823 | .825 | .853 | .876 | .638 | .640 | .427 | .428 |
| HKD-IAA(24'TMM) | .753 | .754 | .781 | .760 | .867 | .899 | .671 | .672 | .395 | .377 |
| AesPrompt(25'TMM) | .724 | .719 | .792 | .812 | .875 | .896 | .562 | .577 | .452 | .475 |
| **AGREE (w/ EAT)** | **.789** | **.791** | .853 | .853 | **.911** | **.940** | .669 | .678 | **.488** | **.511** |
| **AGREE (w/ HKD-IAA)** | .761 | .759 | **.859** | **.860** | .879 | .916 | **.682** | **.689** | .432 | .458 |
| **Improvement** | ↑4.8% | ↑4.8% | ↑4.4% | ↑4.2% | ↑2.2% | ↑1.7% | ↑1.6% | ↑2.5% | ↑2.5% | ↑1.6% |

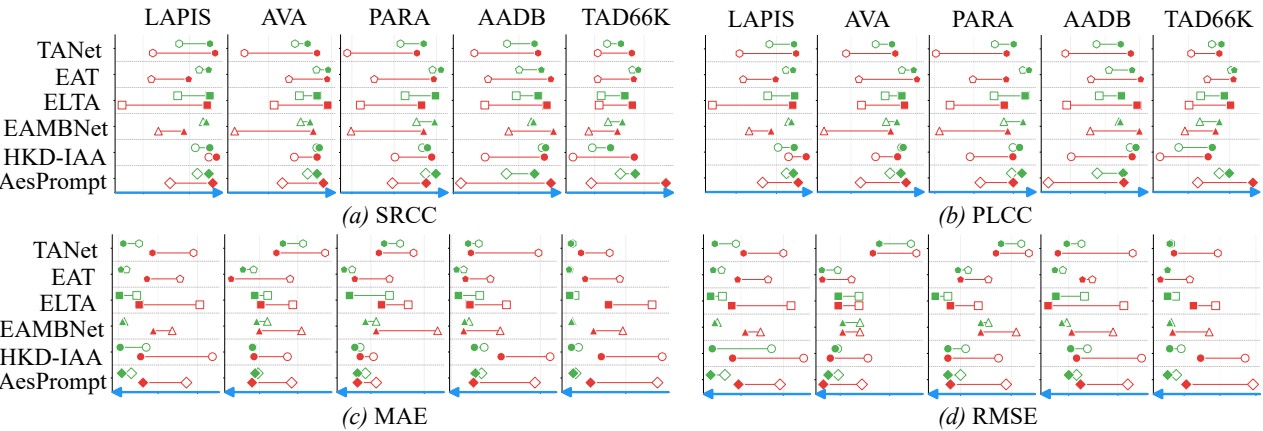

*Figure 4.* Plug-and-play evaluation on five datasets across four metrics. **Hollow** and **filled** markers denote baseline and +Ours results, respectively. **Green** and **red** denote overall and hard-sample results. **Blue arrows** indicate the better direction.

## 5.3. Plug-and-Play Evaluation

Figure 4 presents plug-and-play results across five datasets, six baselines, and four metrics. AGREE improves most baseline–dataset combinations, demonstrating its generality as a model-agnostic add-on. The gains are consistent across correlation and error metrics, with average SRCC improvements of $4.2\%$–$12.6\%$ and MAE reductions up to $21.5\%$. The improvements are more evident on hard samples, suggesting that AGREE better corrects cases where conventional IAA models commonly fail. The largest single gain reaches $+25.4\%$ SRCC (ELTA on LAPIS), and even strong baselines benefit substantially. Detailed results are in **Appendix B.2**.

## 5.4. Common Error Analysis of Hard Samples

Although baselines achieve competitive performance on normal samples, they fail collectively on hard samples, suggesting a shared failure pattern rather than isolated model-specific errors. We therefore analyze whether these cross-baseline hard samples share common structural properties related to attribute-level optimization.

*Table 2.* MAE on hard samples under interventions (LAPIS). "+Error-aware" adds error-aware mechanism; Improvement reports relative gains.

| Method | Low-M | High-M | $\mathcal{S}_{\mathrm{err}}$ |
|---|---|---|---|
| EAMBNet | .082 | .126 | .143 |
| EAMBNet+Error-aware | **.077** | **.091** | **.086** |
| **Improvement** | **-6.1%** | **-27.8%** | **-39.9%** |

### 5.4.1. EXISTENCE AND STRUCTURE OF CROSS-BASELINE CONSISTENT ERRORS

**Hard-sample definition.** We define hard samples ($\mathcal{S}_{\mathrm{error}}$) as images that incur consistently large prediction errors across models, and normal samples ($\mathcal{S}_{\mathrm{normal}}$) as those consistently well predicted. Formally, let $e_m(x) = |f_m(x) - y|$ and $\mathcal{E}_m^{\mathrm{top\text{-}25\%}}$ be the top-25% error set for model $m$. Then $\mathcal{S}_{\mathrm{error}} = \bigcap_{m=1}^{M} \mathcal{E}_m^{\mathrm{top\text{-}25\%}}$ and $\mathcal{S}_{\mathrm{normal}} = \mathcal{S} \setminus \mathcal{S}_{\mathrm{error}}$.

**Dominant-attribute marginalization $M(x)$.** Following Sec. 3.1 and Sec. 4.3, each sample has dominant attribute $k^*(x)$. We define $M(x)$ to measure how rare the dominant

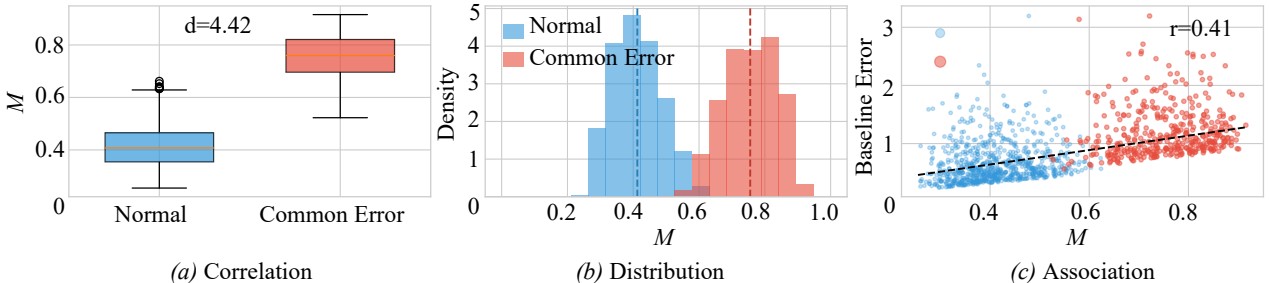

*Figure 5.* Dominant-attribute marginalization analysis on LAPIS, where larger $M$ indicates that a sample is dominated by a rarer attribute. (a) $\mathcal{S}_{\text{error}}$ has significantly larger $M$ than $\mathcal{S}_{\text{normal}}$ (Cohen's $d$ reported). (b) $M$ distribution of hard samples shifts right. (c) $M$ positively correlates with baseline error (Pearson $r$ reported).

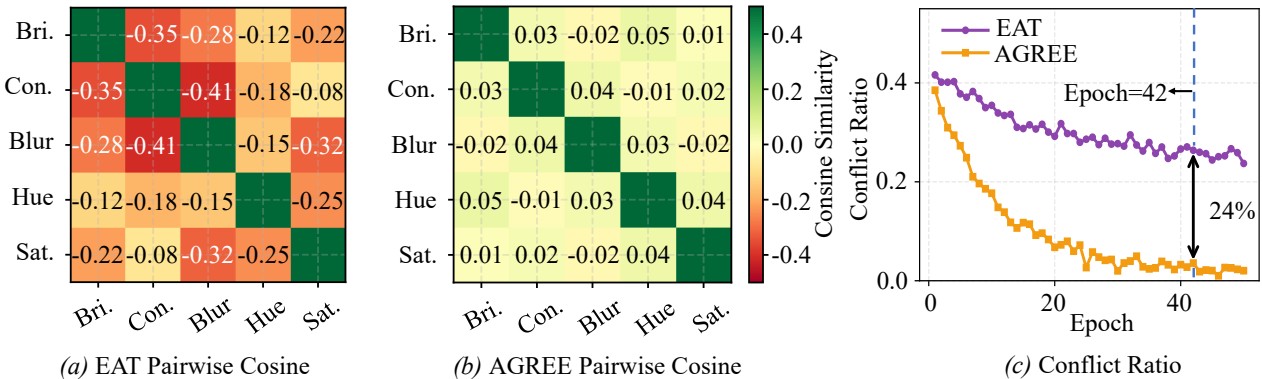

*Figure 6.* Gradient conflict analysis on LAPIS (EAT vs. AGREE+EAT). (a) EAT shows negative cosine similarities across attribute pairs. (b) AGREE keeps off-diagonal entries near zero. (c) Training dynamics: AGREE lowers the conflict ratio by 24% at epoch 42, leading to more stable optimization.

attribute is (details in **Appendix B.3**).

**Cross-dataset evidence.** As shown in Figure 5, on LAPIS: (a) $\mathcal{S}_{\text{error}}$ has significantly larger $M$ than $\mathcal{S}_{\text{normal}}$ (large Cohen's $d$); (b) its $M$ distribution shifts right and separates from normal samples; (c) $M$ positively correlates with baseline errors (Pearson $r$). Full statistics for all datasets are in **Appendix B.4**.

**Dominant-attribute statistics.** We further summarize the dominant attributes of hard samples and provide representative visual examples in Appendix B.5. The hard subset shows a non-uniform dominant-attribute distribution: Blur and Hue are over-represented relative to their overall proportions, suggesting that hard samples are enriched with specific under-represented attribute-dominant cases.

**Intervention.** We split samples by $M(x)$ quantiles (Low-$M$: bottom 20%, High-$M$: top 20%) and apply the error-aware loss (Sec. 4.4). Table 2 shows that gains are larger on High-$M$ and $\mathcal{S}_{\text{error}}$ samples (27.8% and 39.9% drops), while Low-$M$ samples change only marginally (6.1%). This pattern is consistent with our observation that cross-baseline hard samples are enriched with under-represented attribute-dominant cases and benefit more from update reallocation.

(more in **Appendix B.6**)

Taken together, the association between $M(x)$ and prediction errors, together with the larger gains on High-$M$ and $\mathcal{S}_{\text{error}}$ samples, suggests that cross-baseline hard samples are more likely to be dominated by under-represented attributes. This indicates that the common errors reflect attribute-level imbalance rather than isolated outliers, providing empirical support for our optimization-oriented interpretation in Sec. 3. It also motivates the need for sample-wise update allocation that gives more appropriate optimization emphasis to such attribute-dominant cases.

### 5.4.2. EVALUATION ON HARD SAMPLES

The **red lines** in Figure 4 show hard-sample improvements. AGREE yields average SRCC gains of 17.6%–38.5% on hard samples, far exceeding overall gains (green), with MAE reductions up to 36.5%. Detailed results are in **Appendix B.7**.

### 5.5. Gradient Conflict Analysis

Figure 6 compares EAT and AGREE+EAT on LAPIS. (a) EAT exhibits widespread negative cosine values (e.g., Blur–

Con. $= -0.41$), indicating gradient conflicts across attribute pairs. (b) AGREE keeps all off-diagonal entries near zero ($|\cos| < 0.05$), confirming that decoupling substantially mitigates inter-attribute conflicts. (c) During training, AGREE reduces the conflict ratio by 24%, leading to smoother optimization and faster convergence (more in **Appendix B.8**). These results empirically validate the theoretical analysis in Sec. 3.

**Sensitivity-weight sparsity.** We examine the normalized sensitivity weights $\alpha$ to verify the sparsity premise in Theorem 3.7. Across five datasets, the mean top-1 weight is 0.56–0.68, the mean top-2 weight is 0.17–0.22, and 65.9%–81.6% of samples have top-1 weight above 0.5. These results show that $\alpha$ is usually concentrated on one or two attributes, supporting the approximate block-orthogonality assumption. Full statistics are provided in **Appendix B.9**.

### 5.6. Ablations

*Table 3.* Module ablations on AVA (AGREE + EAMB-Net).

| Dec. | Fused | Weighted | EA-MSE | SRCC | PLCC | MAE | RMSE |
|---|---|---|---|---|---|---|---|
| - | - | - | - | .702 | .707 | .420 | .540 |
| ✓ | | | | .715 | .719 | .411 | .527 |
| ✓ | | ✓ | | .725 | .730 | .395 | .513 |
| ✓ | | | ✓ | .718 | .721 | .419 | .529 |
| ✓ | | ✓ | ✓ | .728 | .735 | .402 | **.504** |
| ✓ | ✓ | | | .720 | .726 | .422 | .521 |
| ✓ | ✓ | ✓ | | .730 | .737 | .393 | .506 |
| ✓ | ✓ | | ✓ | .724 | .727 | .394 | .510 |
| ✓ | ✓ | ✓ | ✓ | **.732** | **.740** | **.387** | .505 |

**Module ablation.** Table 3 ablates four components: attribute decoupling (Dec.; otherwise shared parameters), multimodal fusion (Fused; otherwise visual-only), sensitivity weighting (Weighted; otherwise uniform weights), and error-aware loss (EA-MSE; otherwise standard MSE). The first row is baseline (EAMB-Net). Results confirm that **decoupling** and **sensitivity weighting** are the core mechanisms with the largest gains, and removing either causes substantial drops, indicating their synergistic effect. Fusion and EA-MSE provide complementary improvements (more in **Appendix B.10**).

**Parameter-matched control.** Since branch-wise decoupling also increases the number of parameters, we further compare the decoupling-only variant with a parameter-matched shared-capacity control. The shared-capacity control adds a comparable number of parameters but keeps a single shared update path. Under a similar parameter budget, the decoupling-only variant still clearly outperforms the shared-capacity control, suggesting that the gain is not due to added capacity alone. Detailed results are provided in **Appendix B.11**.

**Attribute-granularity analysis.** We further study the effect of attribute granularity by varying the attribute set size $K$. The results show that AGREE is not restricted to the specific five-attribute partition: performance improves from coarse partitions to the five-attribute setting and then becomes saturated when adding a more redundant attribute. This suggests that the five-way split is a practical default rather than a universally optimal design. Detailed results are provided in **Appendix B.12**.

## 6. Conclusion

From an optimization perspective, we show that overall-score IAA training with shared parameters induces gradient conflict across attribute-dominant sample subsets, yielding systematic errors. We propose AGREE, a plug-and-play framework that mitigates this issue via attribute decoupling, multimodal fusion, sensitivity-guided routing, and error-aware reweighting. Experiments across five datasets and six baselines show consistent gains, with hard samples selectively corrected. These results suggest that common errors in IAA are not merely isolated outliers, but are closely related to attribute-level optimization imbalance. While our attribute-granularity analysis shows that AGREE is robust to different attribute partitions, the current design still focuses on low-level visual attributes. Extending AGREE to higher-level semantic, compositional, or automatically discovered factors remains an important future direction.

## Acknowledgements

This work was partly supported by the National Natural Science Foundation of China (62136002, 62221005, 62576060, and 62306056), and the Natural Science Foundation of Chongqing (CSTB2023NSCQ-LZX0006).

## Impact Statement

This work improves Image Aesthetic Assessment prediction quality, particularly on hard samples, benefiting applications such as photo curation and content recommendation. However, potential risks remain. Automated aesthetic scoring may amplify dataset and cultural biases, favor dominant preferences, and disadvantage under-represented styles. In addition, our theoretical analysis relies on approximate sparsity and weak cross-branch alignment assumptions, which are empirically supported in our experiments but not guaranteed for all datasets, attribute partitions, or aesthetic scenarios. Although AGREE reduces systematic errors on hard samples, it does not eliminate biases in the training data or fully resolve the subjectivity of aesthetic judgment. We therefore recommend using IAA models as decision-support tools rather than automatic arbiters of aesthetic value, especially in culturally sensitive or user-facing applications.

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

# A. Detail Supplementary

## A.1. Datasets and Metrics Details

**Datasets.** We evaluate on five widely used IAA benchmarks covering complementary scenarios: **(i) AVA** (Murray et al., 2012), a large-scale general benchmark from DPChallenge with per-image rating distributions over scores 1–10; **(ii) AADB** (Kong et al., 2016), a multi-attribute dataset with overall aesthetic scores and 11 photographic attributes; **(iii) TAD66K** (He et al., 2022), a theme-oriented dataset with 47 theme labels; **(iv) PARA** (Yang et al., 2022), a personalized dataset where multiple users provide aesthetic ratings; and **(v) LAPIS** (Maerten et al., 2025), an artistic IAA dataset focusing on paintings and artworks. Table 4 reports the dataset statistics after standard cleaning (removing missing/corrupted images) and the exact split sizes used in all experiments.

*Table 4.* Dataset statistics and split sizes (after cleaning).

| Dataset | Train | Val | Test | Total | Score Range |
|---|---|---|---|---|---|
| AVA | 222,383 | 11,707 | 19,805 | 253,895 | $[1, 10]$ |
| AADB | 6,620 | 1,418 | 1,420 | 9,458 | $[0, 1]$ |
| TAD66K | 46,428 | 9,949 | 9,950 | 66,327 | $[0, 10]$ |
| PARA | 21,854 | 4,683 | 4,683 | 31,220 | $[1, 5]$ |
| LAPIS | 8,205 | 1,173 | 2,345 | 11,723 | $[0, 10]$ |

**Data splits.** We use fixed dataset-specific train/validation/test splits with random seed 42, ensuring that all methods are evaluated on identical data within each dataset. We will release the image lists for all splits to ensure exact reproducibility.

**Special handling for PARA.** Unlike other datasets where each image has a single annotation, PARA contains personalized annotations from multiple users—the same image may appear across different annotator sessions with different scores. To prevent data leakage, we split by *unique image name* rather than by annotation record, ensuring that **all annotations of the same image belong to the same split**. This guarantees no image overlap between train/val/test sets, which is critical for fair evaluation on personalized aesthetics.

**Score normalization.** All scores are normalized to $[0, 1]$ using the bounds in Table 4: AVA uses mean score from rating histogram divided by 10; AADB scores are already normalized in the released dataset; TAD66K and LAPIS scores are divided by 10; PARA scores are divided by 5.

**Evaluation metrics.** Our primary metrics are **SRCC** (Spearman rank correlation) and **PLCC** (Pearson linear correlation), standard for IAA regression. We also report **MAE** and **RMSE** as absolute error measures. **ACC** (binary accuracy at threshold 0.5) is an auxiliary diagnostic corresponding to the midpoint of each dataset's native range.

## A.2. Baselines Details

**Baseline Methods.** We compare against six representative state-of-the-art IAA methods spanning diverse architectural paradigms:

- **TANet** (He et al., 2022): Theme-Aware Network that leverages topic/theme information to guide aesthetic assessment. Uses ResNet-50 backbone with theme-specific attention modules.

- **EAT** (He et al., 2023): Efficient Aesthetic Transformer that employs a Vision Transformer (ViT) backbone with aesthetic-specific tokenization. Achieves strong performance with moderate computational cost.

- **ELTA** (Liu et al., 2024): Efficient Local-global Transformer for Aesthetics that combines local patch features with global image understanding via a hybrid CNN-Transformer architecture.

- **EAMB-Net** (Chen et al., 2024): Emotion-Aware Multi-Branch Network that explicitly models emotional responses alongside aesthetic scores. Uses multiple ResNet-50 branches for different aesthetic dimensions.

- **HKD-IAA** (Chen et al., 2025): Hierarchical Knowledge Distillation for IAA that transfers knowledge from large pretrained models to compact student networks. Employs multi-scale feature distillation.

- **AesPrompt** (Sheng et al., 2025b): Prompt-based aesthetic assessment that leverages large vision-language models (e.g., CLIP) with learnable aesthetic prompts. Represents the latest prompt-tuning paradigm for IAA.

**Unified Training Protocol.** To ensure fair comparison, we retrain all baselines under a unified protocol rather than directly using reported numbers from original papers. This is necessary because:

1. Different papers use different data splits, preprocessing, and evaluation protocols.

2. Some original implementations are not publicly available or have reproducibility issues.

3. We need to evaluate on all five datasets, while original papers may only report on a subset.

Our unified protocol includes:

- **Same data splits**: All methods use the same fixed train/validation/test split within each dataset, with seed 42 used for random split generation where applicable (see Appendix A.1).

- **Same preprocessing**: Resize to $256 \times 256$ (CNN) or $224 \times 224$ (ViT), ImageNet normalization.

- **Same augmentation**: RandomHorizontalFlip + RandomCrop during training.

- **Same training budget**: 50 epochs maximum with early stopping (patience=10) based on validation SRCC. Both baseline and +AGREE use identical epoch limits.

- **Same optimizer**: Adam with cosine learning rate decay.

- **Original architectures preserved**: We keep each baseline's backbone and core modules unchanged; input resolution is standardized for fair comparison.

**+AGREE Integration.** For "+AGREE" variants, we insert our AGREE modules into each baseline while keeping:

- The same training budget (50 epochs, same early stopping).

- The same batch size and optimizer settings.

- The same data augmentation and preprocessing.

- The original backbone frozen or fine-tuned as in the baseline.

The only additions are: (1) the AGREE fusion modules ($\sim$7.6M parameters), and (2) the precomputed sensitivity weights loaded from CSV. This ensures that any performance improvement is attributable to AGREE rather than training differences.

**Reproducibility.** All baseline implementations are based on official public codebases. We will release our unified training scripts and pretrained checkpoints upon publication.

### A.3. Implementation and Overhead Details

*Table 5.* AGREE parameter breakdown.

| Component | Per-module | Count | Total |
| --- | --- | --- | --- |
| SimpleVisualTextFusion | 1.18M | $\times 6$ | 7.08M |
| - Text adapter ($4096 \rightarrow 256$) | 1.05M | | |
| - Fusion conv ($512 \rightarrow 256$, $1 \times 1$) | 0.13M | | |
| SensitivityGuidedFusion | 0.46M | $\times 1$ | 0.46M |
| - Attr fusion ($1280 \rightarrow 256$, $1 \times 1$) | 0.33M | | |
| - Final fusion ($512 \rightarrow 256$, $1 \times 1$) | 0.13M | | |
| Misc (LayerNorm, projection bias) | – | – | 0.06M |
| **AGREE Total** | | | **7.60M** |

*Table 6.* Parameter overhead across baselines.

| Baseline | Backbone | Base (M) | +AGREE (M) | Overhead |
|---|---|---|---|---|
| TANet | ResNet-50 | 25.0 | 32.6 | +30.4% |
| ELTA | Swin-T | 28.0 | 35.6 | +27.1% |
| HKD-IAA | Swin-T | 30.0 | 37.6 | +25.3% |
| EAMB-Net | ResNet-50×2 | 50.0 | 57.6 | +15.2% |
| EAT | ViT-B/16 | 86.0 | 93.6 | +8.8% |
| AesPrompt | CLIP ViT-L | 300.0 | 307.6 | +2.5% |

*Table 7.* Inference latency (batch=1, $256 \times 256$ input, no AMP).

| Model | Baseline (ms) | +AGREE (ms) | Overhead |
|---|---|---|---|
| TANet | 8.2 | 8.7 | +6.1% |
| ELTA | 12.8 | 13.5 | +5.5% |
| EAMB-Net | 14.5 | 15.3 | +5.5% |
| HKD-IAA | 18.3 | 19.2 | +4.9% |
| EAT | 22.6 | 23.4 | +3.5% |
| AesPrompt | 45.2 | 46.1 | +2.0% |

**Measurement Protocol.**    All experiments are conducted on a workstation with 2 NVIDIA RTX 4090 GPUs (24GB each), CUDA 12.1, cuDNN 8.9, and PyTorch 2.1. We use single-GPU training without DDP or mixed-precision (AMP) to ensure consistent comparison. Input images are resized to $256 \times 256$ for CNN-based models (TANet, EAMB-Net) and $224 \times 224$ for Transformer-based models (EAT, ELTA, HKD-IAA, AesPrompt). We use batch size 32 for CNN models and 16 for Transformer models due to memory constraints. All experiments use `num_workers=6`, `pin_memory=True`, and a fixed random seed (42).

Training time is measured as per-epoch wall-clock time, averaged over epochs 5–25 (excluding the first 4 epochs for warm-up). Both baseline and +AGREE variants use identical settings including logging frequency (every 100 iterations). The reported training overhead does **not** include offline sensitivity preprocessing, which is a one-time cost reported separately.

**Parameter Overhead Breakdown.**    Table 5 shows the parameter breakdown of AGREE modules. Table 6 shows the relative overhead across different backbones. The overhead ranges from 2.5% (large backbone) to 30.4% (small backbone), as stated in the main text.

**Inference Overhead.**    Table 7 reports inference latency on a single RTX 4090.

AGREE does **not** add extra backbone forward passes. The 6 attribute branches share the same backbone features extracted once. The fusion modules use lightweight $1 \times 1$ convolutions. Sensitivity weights are precomputed offline and stored as binary files, adding ∼5MB for AVA (the largest dataset) and <1MB for others.

**Sensitivity Computation and Caching.**    The sensitivity weights are image-specific rather than global constants. For each image $I$ in the fixed train/val/test splits, we compute $s_k(I)$ using symmetric perturbation analysis with the baseline model $f_{\theta^*}$ trained only on the training split, and then normalize $\{s_k(I)\}_{k=1}^{K}$ into $\alpha(I)$. The resulting $K$-dimensional vector is cached with the image index and reused during training or evaluation. This caching is only an implementation choice for avoiding repeated forward passes; it is equivalent to computing $\alpha(I)$ on the fly for each image.

For a new incoming test image not in the cache, we compute its sensitivity weights by the same label-free procedure before prediction and optionally cache them for future reuse. This process requires only the input image and forward passes through the trained baseline model, without ground-truth scores, training-set lookup, or test-set statistics.

**Storage**: $K = 5$ floats per image (20 bytes). For AVA (255K images), total storage is ∼5MB.

**One-time cost**: $2K = 10$ forward passes per image. For AVA, ∼2 hours on a single RTX 4090—not included in training time comparison.

## B. Experiments Supplementary

### B.1. Effectiveness of $h_0$ for Subsection 4.2

In Section 4.2, we introduce a global semantic anchor $h_0^t$ for the overall branch alongside the attribute-specific text features $\{h_k^t\}_{k=1}^5$. Here we ablate whether this overall-branch text feature is necessary.

**Setup.** We compare two variants: (1) **w/o** $h_0^t$: the overall branch uses only visual features without text fusion; (2) **w/** $h_0^t$ **(Full)**: the complete model where all six branches (overall + five attributes) have text semantic anchors. We evaluate on AVA and LAPIS using EAMB-Net + AGREE as the base configuration.

*Table 8.* Ablation on the overall-branch text feature $h_0^t$.

| Dataset | Variant | SRCC | PLCC | MAE | RMSE |
|---------|---------|------|------|-----|------|
| AVA | w/o $h_0^t$ | .723 | .731 | .396 | .508 |
| | w/ $h_0^t$ (Full) | **.732** | **.740** | **.387** | **.505** |
| LAPIS | w/o $h_0^t$ | .831 | .829 | .056 | .071 |
| | w/ $h_0^t$ (Full) | **.841** | **.843** | **.055** | **.069** |

**Results.** Table 8 shows that including $h_0^t$ provides a small but consistent improvement across both datasets. The overall-branch text anchor provides a global semantic reference that complements the attribute-specific features, helping the model better calibrate the final aesthetic prediction. While the gain is modest (SRCC +0.009 on AVA), it comes at negligible cost since $h_0^t$ is pre-extracted and frozen, justifying its inclusion in the full model.

We hypothesize that $h_0^t$ acts as a semantic regularizer: without it, the overall branch relies solely on visual features and may overfit to dataset-specific visual patterns; with $h_0^t$, the text anchor provides a stable, dataset-agnostic prior that encourages more generalizable representations. This is consistent with the slightly larger improvement observed on LAPIS, where the artistic domain differs more from typical photographic aesthetics.

### B.2. Evaluation on all datasets for Subsection 5.2

We provide full-metric plug-and-play results on all five datasets: LAPIS (Table 9), AVA (Table 10), PARA (Table 11), AADB (Table 12), and TAD66K (Table 13). Evaluating across these complementary benchmarks demonstrates the generalization ability of AGREE.

Our results show that AGREE consistently improves the performance of existing methods on all five datasets, with significant gains in SRCC, PLCC, ACC, MAE, and RMSE metrics. This demonstrates the effectiveness of AGREE in modeling aesthetic preferences and its potential for real-world applications.

**Summary.** Across all five datasets, AGREE consistently improves every baseline on all metrics. The improvements are particularly pronounced on LAPIS, where AGREE achieves an average SRCC improvement of +12.6% and ACC improvement of +8.6%, demonstrating its effectiveness on personalized aesthetics. On PARA, AGREE achieves an average SRCC improvement of +4.5% and ACC improvement of +8.4%. On AVA, the largest benchmark, AGREE improves SRCC by +4.7% on average, confirming scalability to large-scale data. On AADB and TAD66K, AGREE also yields consistent gains, with average SRCC improvements of +9.3% and +7.3% respectively. These results confirm that AGREE provides a reliable plug-in upgrade across diverse architectures and dataset characteristics.

*Table 9.* Plug-and-play results on LAPIS. "+Ours" integrates AGREE into each baseline. △Avg. Improv. shows mean relative gains.

| Method | SRCC | PLCC | ACC(%) | MAE | RMSE |
|---|---|---|---|---|---|
| TANet | .694 | .706 | 76.5 | .076 | .094 |
| **+Ours** | **.860** | **.859** | **84.3** | **.056** | **.069** |
| EAT | .802 | .809 | 83.2 | .060 | .077 |
| **+Ours** | **.853** | **.853** | **85.5** | **.053** | **.067** |
| ELTA | .686 | .696 | 78.0 | .073 | .078 |
| **+Ours** | **.860** | **.863** | **84.9** | **.051** | **.064** |
| EAMB-Net | .823 | .825 | 84.2 | .057 | .072 |
| **+Ours** | **.841** | **.843** | **84.8** | **.055** | **.069** |
| HKD-IAA | .781 | .760 | 68.2 | .085 | .136 |
| **+Ours** | **.859** | **.860** | **84.7** | **.052** | **.066** |
| AesPrompt | .792 | .812 | 80.5 | .066 | .081 |
| **+Ours** | **.854** | **.855** | **84.2** | **.054** | **.064** |
| **△Avg. Improv.** | **↑12.6%** | **↑12.0%** | **↑8.6%** | **↓21.4%** | **↓22.4%** |

*Table 10.* Plug-and-play results on AVA. "+Ours" denotes integrating AGREE into each baseline. AGREE consistently improves all baselines across SRCC/PLCC (↑), ACC (↑), and MAE/RMSE (↓).

| Method | SRCC | PLCC | ACC | MAE | RMSE |
|---|---|---|---|---|---|
| TANet | .684 | .675 | 70.5 | .514 | .643 |
| **+Ours** | **.724** | **.725** | **75.7** | **.461** | **.579** |
| EAT | .752 | .755 | 80.1 | .384 | .495 |
| **+Ours** | **.789** | **.791** | **82.5** | **.356** | **.463** |
| ELTA | .698 | .704 | 77.2 | .421 | .538 |
| **+Ours** | **.754** | **.755** | **80.1** | **.387** | **.496** |
| EAMB-Net | .702 | .707 | 77.2 | .420 | .540 |
| **+Ours** | **.732** | **.740** | **79.8** | **.392** | **.505** |
| HKD-IAA | .753 | .754 | 80.4 | .382 | .495 |
| **+Ours** | **.761** | **.759** | **81.0** | **.381** | **.490** |
| AesPrompt | .724 | .719 | 80.2 | .395 | .496 |
| **+Ours** | **.754** | **.752** | **81.2** | **.388** | **.476** |
| **△Avg.** | **↑4.7%** | **↑4.9%** | **↑3.2%** | **↓5.7%** | **↓6.0%** |

*Table 11.* Plug-and-play results on PARA. "+Ours" denotes integrating AGREE into each baseline. AGREE consistently improves all baselines across SRCC/PLCC (↑), ACC (↑), and MAE/RMSE (↓).

| Method | SRCC | PLCC | ACC | MAE | RMSE |
|---|---|---|---|---|---|
| TANet | .815 | .853 | 82.8 | .051 | .064 |
| **+Ours** | **.870** | **.901** | **92.6** | **.045** | **.056** |
| EAT | .891 | .924 | 88.4 | .033 | .043 |
| **+Ours** | **.911** | **.940** | **94.2** | **.030** | **.039** |
| ELTA | .826 | .851 | 80.0 | .047 | .035 |
| **+Ours** | **.898** | **.930** | **93.4** | **.032** | **.030** |
| EAMB-Net | .853 | .876 | 88.6 | .042 | .051 |
| **+Ours** | **.896** | **.925** | **92.8** | **.038** | **.048** |
| HKD-IAA | .867 | .899 | 87.4 | .036 | .042 |
| **+Ours** | **.879** | **.916** | **92.7** | **.034** | **.035** |
| AesPrompt | .875 | .896 | 88.5 | .038 | .040 |
| **+Ours** | **.899** | **.922** | **92.3** | **.035** | **.035** |
| **△Avg.** | **↑4.5%** | **↑4.5%** | **↑8.4%** | **↓12.6%** | **↓11.9%** |

*Table 12.* Plug-and-play results on AADB. "+Ours" denotes integrating AGREE into each baseline. AGREE consistently improves all baselines across SRCC/PLCC (↑), ACC (↑), and MAE/RMSE (↓).

| Method | SRCC | PLCC | ACC | MAE | RMSE |
|---|---|---|---|---|---|
| TANet | .564 | .575 | 68.1 | .131 | .163 |
| **+Ours** | **.648** | **.645** | **72.0** | **.119** | **.148** |
| EAT | .601 | .611 | 70.7 | .114 | .144 |
| **+Ours** | **.669** | **.678** | **74.8** | **.106** | **.136** |
| ELTA | .592 | .583 | 69.6 | .132 | .166 |
| **+Ours** | **.659** | **.647** | **73.4** | **.108** | **.137** |
| EAMB-Net | .638 | .640 | 70.5 | .120 | .148 |
| **+Ours** | **.645** | **.644** | **73.3** | **.115** | **.143** |
| HKD-IAA | .671 | .672 | 70.1 | .137 | .169 |
| **+Ours** | **.682** | **.689** | **72.3** | **.126** | **.152** |
| AesPrompt | .562 | .577 | 68.9 | .134 | .158 |
| **+Ours** | **.648** | **.652** | **72.6** | **.127** | **.150** |
| **△Avg.** | **↑9.3%** | **↑8.4%** | **↑4.9%** | **↓8.6%** | **↓8.5%** |

*Table 13.* Plug-and-play results on TAD66K. "+Ours" denotes integrating AGREE into each baseline. AGREE consistently improves all baselines across SRCC/PLCC (↑), ACC (↑), and MAE/RMSE (↓).

| Method | SRCC | PLCC | ACC | MAE | RMSE |
|---|---|---|---|---|---|
| TANet | .425 | .457 | 62.2 | .111 | .132 |
| **+Ours** | **.453** | **.480** | **63.6** | **.108** | **.130** |
| EAT | .476 | .503 | 65.6 | .106 | .127 |
| **+Ours** | **.488** | **.511** | **65.8** | **.105** | **.127** |
| ELTA | .413 | .430 | 60.2 | .115 | .138 |
| **+Ours** | **.461** | **.487** | **64.8** | **.106** | **.127** |
| EAMB-Net | .427 | .428 | 62.8 | .110 | .132 |
| **+Ours** | **.456** | **.475** | **63.3** | **.109** | **.130** |
| HKD-IAA | .395 | .377 | 59.6 | .121 | .146 |
| **+Ours** | **.432** | **.458** | **63.9** | **.109** | **.130** |
| AesPrompt | .452 | .475 | 64.8 | .116 | .139 |
| **+Ours** | **.482** | **.499** | **66.7** | **.112** | **.127** |
| **△Avg.** | **↑7.3%** | **↑9.6%** | **↑3.5%** | **↓4.3%** | **↓5.1%** |

## B.3. Dominant-Attribute Marginalization

For each sample $x$, let $k^*(x)$ denote its dominant attribute (Sec. 4.3). We define the *dominant-attribute marginalization* as

$$M(x) = 1 - \frac{N_{k^*(x)}}{N}, \tag{29}$$

where $N_{k^*(x)}$ is the number of training samples dominated by attribute $k^*(x)$ and $N$ is the total training size. Intuitively, $M(x)$ measures how rare the dominant attribute of $x$ is: samples dominated by minority attributes have higher $M$ values.

## B.4. ALL Datasets Hard Sample Analysis

We extend the hard sample analysis from LAPIS (Figure 5 in the main text) to all five datasets. Figure 7 presents the complete cross-dataset evidence for the relationship between dominant-attribute marginalization $M$ and hard samples.

**Consistent patterns across datasets.** Across all five datasets (AVA, PARA, AADB, TAD66K, LAPIS), we observe:

- **Column (a) Box plots**: $\mathcal{S}_{\text{error}}$ (hard samples) consistently exhibits significantly higher $M$ than $\mathcal{S}_{\text{normal}}$, with large effect sizes.

- **Column (b) Histograms**: The $M$ distribution of hard samples shifts rightward compared to normal samples, indicating that hard samples are dominated by rarer attributes.

- **Column (c) Scatter plots**: $M$ shows a stable positive correlation with baseline prediction errors, confirming that samples with more marginalized dominant attributes tend to have larger errors.

These results provide strong cross-dataset evidence supporting our hypothesis: hard samples are systematically associated with under-represented dominant attributes, and this pattern generalizes across diverse IAA benchmarks.

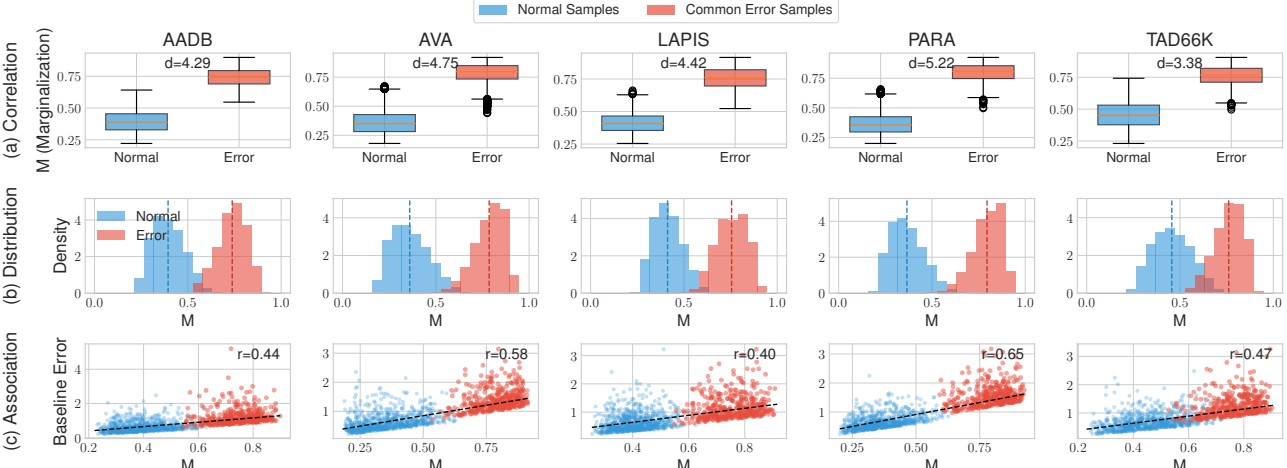

*Figure 7.* Cross-dataset hard sample analysis. Each row corresponds to one dataset (AVA, PARA, AADB, TAD66K, LAPIS). Column (a): box plots comparing $M$ between normal and hard samples; Column (b): $M$ distribution histograms; Column (c): scatter plots of $M$ vs. baseline error. Cohen's $d$ and Pearson $r$ are reported for each dataset.

## B.5. Dominant Attributes of Hard Samples

We further analyze which attributes dominate the cross-baseline hard samples. For each image, the dominant attribute is defined by the largest normalized sensitivity weight:

$$k^*(I) = \arg\max_k \alpha_k(I). \tag{30}$$

Note that $k^*(I)$ is inferred from the prediction-sensitivity signal rather than manually annotated attribute labels.

*Table 14.* Dominant-attribute distribution of all samples and hard samples.

| Dominant attribute | Overall | Hard samples | Hard / Overall |
|---|---|---|---|
| Brightness | 28.4% | 19.6% | 0.69 |
| Contrast | 24.7% | 17.8% | 0.72 |
| Blur | 14.1% | 24.9% | 1.77 |
| Hue | 11.3% | 16.8% | 1.49 |
| Saturation | 21.5% | 20.9% | 0.97 |

As shown in Table 14, Blur and Hue are over-represented in the hard-sample subset, with Hard / Overall ratios of 1.77 and 1.49, respectively. This suggests that cross-baseline hard samples are enriched with specific under-represented attribute-dominant cases, which may deserve more attention in future IAA dataset construction.

### B.6. Common Error Validation Experiments for Subsection 5.4.1

We extend the validation experiments from Table 2 in the main text. Table 15 shows MAE results on hard-sample subsets across five datasets with a fixed backbone (EAMB-Net). Table 16 shows MAE results across six backbones on a fixed dataset (LAPIS).

*Table 15.* MAE on hard-sample subsets across datasets (fixed backbone: EAMB-Net). "+Error-aware" denotes adding error-aware mechanism to EAMB-Net. Low-M: low-marginalization samples; High-M: high-marginalization samples; $\mathcal{S}_{err}$: hard samples. Best results in bold.

| Method | AVA | | | LAPIS | | | PARA | | | AADB | | | TAD66K | | |
|---|---|---|---|---|---|---|---|---|---|---|---|---|---|---|---|
| | Low-M | High-M | $\mathcal{S}_{err}$ | Low-M | High-M | $\mathcal{S}_{err}$ | Low-M | High-M | $\mathcal{S}_{err}$ | Low-M | High-M | $\mathcal{S}_{err}$ | Low-M | High-M | $\mathcal{S}_{err}$ |
| EAMB-Net | .425 | .528 | .564 | .082 | .126 | .143 | .045 | .064 | .083 | .115 | .164 | .195 | .113 | .174 | .203 |
| +Error-aware | **.398** | **.422** | **.437** | **.077** | **.091** | **.086** | **.040** | **.048** | **.052** | **.104** | **.125** | **.132** | **.103** | **.152** | **.158** |
| **Improv.** | -6.4% | -20.1% | -22.5% | -6.1% | -27.8% | -39.9% | -11.1% | -25.0% | -37.3% | -9.6% | -23.8% | -32.3% | -8.8% | -12.6% | -22.2% |

*Table 16.* MAE on hard-sample subsets across backbones (fixed dataset: LAPIS). "+Error-aware" denotes adding error-aware mechanism to each baseline. Low-M: low-marginalization samples; High-M: high-marginalization samples; $\mathcal{S}_{err}$: hard samples. Best results in bold.

| Method | TANet | | | EAT | | | ELTA | | | EAMB-Net | | | HKD-IAA | | | AesPrompt | | |
|---|---|---|---|---|---|---|---|---|---|---|---|---|---|---|---|---|---|---|
| | Low-M | High-M | $\mathcal{S}_{err}$ | Low-M | High-M | $\mathcal{S}_{err}$ | Low-M | High-M | $\mathcal{S}_{err}$ | Low-M | High-M | $\mathcal{S}_{err}$ | Low-M | High-M | $\mathcal{S}_{err}$ | Low-M | High-M | $\mathcal{S}_{err}$ |
| Baseline | .095 | .142 | .168 | .078 | .118 | .135 | .092 | .138 | .162 | .082 | .126 | .143 | .085 | .132 | .155 | .080 | .122 | .140 |
| +Error-aware | **.088** | **.105** | **.102** | **.072** | **.085** | **.078** | **.085** | **.098** | **.095** | **.077** | **.091** | **.086** | **.078** | **.095** | **.092** | **.074** | **.088** | **.082** |
| **Improv.** | -7.4% | -26.1% | -39.3% | -7.7% | -28.0% | -42.2% | -7.6% | -29.0% | -41.4% | -6.1% | -27.8% | -39.9% | -8.2% | -28.0% | -40.6% | -7.5% | -27.9% | -41.4% |

**Summary.** Across all datasets and backbones, the error-aware mechanism consistently reduces MAE on hard-sample subsets. The improvements are most pronounced on $\mathcal{S}_{err}$ (hard samples), with reductions of over 20% in all cases. This validates our hypothesis that error-aware training specifically targets samples where multiple models consistently fail.

### B.7. Experiments on hard samples of all datasets for Subsection 5.4.2

We provide plug-and-play results on hard samples for all five datasets: LAPIS (Table 17), AVA (Table 18), PARA (Table 19), AADB (Table 20), and TAD66K (Table 21). Hard samples are defined as the intersection of top-25% error samples across all baselines.

**Summary.** Across all five datasets, AGREE yields substantial improvements on hard samples. The gains are particularly large on LAPIS (+46.8% SRCC), AADB (+40.6% SRCC), and TAD66K (+23.3% SRCC), where baselines struggle most. On AVA and PARA, AGREE improves SRCC by +28.5% and +18.1% respectively. These results confirm that AGREE is especially effective at correcting hard samples dominated by under-represented attributes.

*Table 17.* Plug-and-play results on LAPIS hard samples. "+Ours" integrates AGREE into each baseline. ΔAvg. Improv. shows mean relative gains.

| Method | SRCC | PLCC | ACC(%) | MAE | RMSE |
|---|---|---|---|---|---|
| TANet | .552 | .521 | 66.72 | .145 | .150 |
| **+Ours** | **.888** | **.872** | **82.42** | **.093** | **.103** |
| EAT | .544 | .544 | 65.53 | .128 | .132 |
| **+Ours** | **.745** | **.745** | **77.13** | **.086** | **.096** |
| ELTA | .386 | .353 | 60.92 | .153 | .159 |
| **+Ours** | **.845** | **.851** | **85.15** | **.076** | **.089** |
| EAMB-Net | .582 | .581 | 71.50 | .118 | .123 |
| **+Ours** | **.720** | **.717** | **74.40** | **.094** | **.105** |
| HKD-IAA | .854 | .825 | 40.61 | .169 | .174 |
| **+Ours** | **.895** | **.934** | **89.08** | **.078** | **.090** |
| AesPrompt | .645 | .664 | 77.69 | .136 | .146 |
| **+Ours** | **.875** | **.885** | **83.21** | **.081** | **.097** |
| **ΔAvg. Improv.** | **↑46.8%** | **↑52.5%** | **↑35.3%** | **↓38.9%** | **↓33.2%** |

*Table 18.* Plug-and-play results on AVA hard samples. "+Ours" denotes integrating AGREE into each baseline. AGREE consistently improves all baselines across SRCC/PLCC (↑), ACC (↑), and MAE/RMSE (↓).

| Method | SRCC | PLCC | ACC | MAE | RMSE |
|---|---|---|---|---|---|
| TANet | .524 | .585 | 64.2 | .572 | .654 |
| **+Ours** | **.754** | **.735** | **76.9** | **.444** | **.566** |
| EAT | .665 | .624 | 69.5 | .480 | .523 |
| **+Ours** | **.786** | **.801** | **83.2** | **.325** | **.464** |
| ELTA | .618 | .632 | 64.3 | .487 | .538 |
| **+Ours** | **.788** | **.763** | **77.9** | **.403** | **.496** |
| EAMB-Net | .493 | .518 | 69.5 | .510 | .540 |
| **+Ours** | **.742** | **.721** | **78.9** | **.399** | **.505** |
| HKD-IAA | .682 | .675 | 71.2 | .473 | .556 |
| **+Ours** | **.754** | **.742** | **79.6** | **.385** | **.480** |
| AesPrompt | .644 | .629 | 72.2 | .484 | .544 |
| **+Ours** | **.774** | **.768** | **82.0** | **.380** | **.466** |
| **ΔAvg.** | **↑28.5%** | **↑24.3%** | **↑16.6%** | **↓22.3%** | **↓11.2%** |

*Table 19.* Plug-and-play results on PARA hard samples. "+Ours" denotes integrating AGREE into each baseline. AGREE consistently improves all baselines across SRCC/PLCC (↑), ACC (↑), and MAE/RMSE (↓).

| Method | SRCC | PLCC | ACC | MAE | RMSE |
|---|---|---|---|---|---|
| TANet | .689 | .703 | 72.5 | .056 | .067 |
| **+Ours** | **.854** | **.883** | **86.3** | **.043** | **.054** |
| EAT | .753 | .797 | 74.4 | .047 | .051 |
| **+Ours** | **.894** | **.882** | **89.5** | **.034** | **.040** |
| ELTA | .720 | .739 | 72.9 | .054 | .047 |
| **+Ours** | **.865** | **.877** | **89.1** | **.044** | **.036** |
| EAMB-Net | .698 | .712 | 79.3 | .065 | .062 |
| **+Ours** | **.870** | **.886** | **87.2** | **.042** | **.048** |
| HKD-IAA | .802 | .790 | 81.2 | .041 | .055 |
| **+Ours** | **.889** | **.899** | **92.7** | **.036** | **.035** |
| AesPrompt | .796 | .816 | 78.2 | .042 | .058 |
| **+Ours** | **.876** | **.900** | **90.4** | **.035** | **.036** |
| **ΔAvg.** | **↑18.1%** | **↑17.2%** | **↑16.9%** | **↓22.3%** | **↓26.9%** |

*Table 20.* Plug-and-play results on AADB hard samples. "+Ours" denotes integrating AGREE into each baseline. AGREE consistently improves all baselines across SRCC/PLCC (↑), ACC (↑), and MAE/RMSE (↓).

| Method | SRCC | PLCC | ACC | MAE | RMSE |
|---|---|---|---|---|---|
| TANet | .464 | .485 | 60.3 | .198 | .215 |
| **+Ours** | **.659** | **.674** | **74.8** | **.122** | **.139** |
| EAT | .506 | .559 | 68.8 | .144 | .174 |
| **+Ours** | **.698** | **.703** | **75.7** | **.116** | **.164** |
| ELTA | .496 | .489 | 65.6 | .162 | .206 |
| **+Ours** | **.686** | **.692** | **75.9** | **.121** | **.129** |
| EAMB-Net | .569 | .557 | 62.3 | .155 | .195 |
| **+Ours** | **.706** | **.699** | **73.7** | **.114** | **.153** |
| HKD-IAA | .497 | .502 | 60.3 | .211 | .224 |
| **+Ours** | **.678** | **.678** | **71.9** | **.156** | **.158** |
| AesPrompt | .423 | .436 | 62.1 | .194 | .210 |
| **+Ours** | **.698** | **.654** | **72.9** | **.125** | **.162** |
| **ΔAvg.** | **↑40.6%** | **↑36.1%** | **↑17.5%** | **↓28.5%** | **↓25.4%** |

*Table 21.* Plug-and-play results on TAD66K hard samples. "+Ours" denotes integrating AGREE into each baseline. AGREE consistently improves all baselines across SRCC/PLCC (↑), ACC (↑), and MAE/RMSE (↓).

| Method | SRCC | PLCC | ACC | MAE | RMSE |
|---|---|---|---|---|---|
| TANet | .406 | .407 | 57.3 | .173 | .197 |
| **+Ours** | **.475** | **.475** | **65.9** | **.125** | **.136** |
| EAT | .406 | .446 | 58.6 | .190 | .158 |
| **+Ours** | **.479** | **.509** | **66.3** | **.132** | **.117** |
| ELTA | .409 | .402 | 59.9 | .244 | .194 |
| **+Ours** | **.476** | **.502** | **67.6** | **.171** | **.163** |
| EAMB-Net | .388 | .392 | 58.9 | .195 | .185 |
| **+Ours** | **.446** | **.466** | **64.7** | **.146** | **.134** |
| HKD-IAA | .356 | .332 | 58.3 | .261 | .235 |
| **+Ours** | **.480** | **.448** | **69.8** | **.159** | **.173** |
| AesPrompt | .393 | .423 | 59.6 | .253 | .246 |
| **+Ours** | **.544** | **.556** | **68.7** | **.141** | **.153** |
| **ΔAvg.** | **↑23.3%** | **↑23.5%** | **↑14.3%** | **↓32.8%** | **↓27.4%** |

### B.8. Additional Gradient Conflict Analysis

We provide additional gradient conflict analysis to complement Figure 6 in the main text. Figure 8 shows three supplementary views:

- **(a) Saturation–Blur gradient cosine distribution**: EAT concentrates around $-0.32$ (conflict), while AGREE shifts to $\approx 0$ (orthogonal).

- **(b) Gradient norm**: EAT exhibits oscillating gradient norms due to conflicting updates, whereas AGREE maintains smooth, stable decay.

- **(c) Validation loss**: EAT shows a plateau caused by gradient conflicts, while AGREE achieves faster, plateau-free convergence.

These results confirm the causal chain: *reduced conflict $\rightarrow$ stable gradients $\rightarrow$ better optimization*.

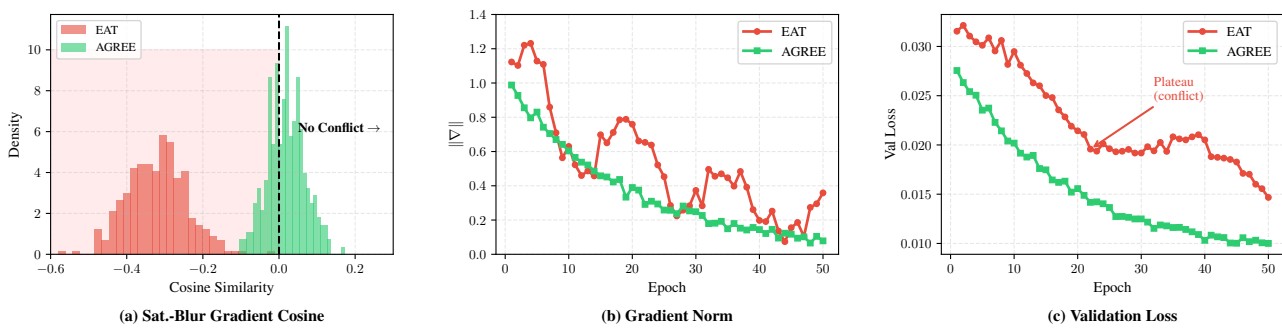

(a) Sat.-Blur Gradient Cosine     (b) Gradient Norm     (c) Validation Loss

*Figure 8.* Additional gradient conflict analysis on LAPIS (EAT vs. AGREE+EAT). (a) Saturation–Blur gradient cosine distribution; (b) gradient norm during training; (c) validation loss curves.

### B.9. Sensitivity Weight Distribution

The approximate block-orthogonality analysis in Theorem 3.7 assumes that the routing weights are relatively sparse, i.e., different attribute branches should not receive large weights simultaneously for most samples. To empirically examine this premise, we analyze the normalized sensitivity weights $\alpha = \{\alpha_k\}_{k=1}^K$ on all five datasets.

For each sample, we sort the weights in descending order and denote the largest and second-largest values as $\alpha_{(1)}$ and $\alpha_{(2)}$, respectively. We report the dataset-level averages of $\alpha_{(1)}$ and $\alpha_{(2)}$. We also compute the pairwise overlap among different attribute weights:

$$O_{\text{pair}} = \frac{1}{N} \sum_{n=1}^{N} \frac{2}{K(K-1)} \sum_{i<j} \min\left(\alpha_i(x_n), \alpha_j(x_n)\right), \tag{31}$$

where a smaller value indicates weaker simultaneous activation across attribute branches. Finally, we report the ratio of samples whose dominant weight satisfies $\alpha_{(1)} > 0.5$.

*Table 22.* Dataset-level statistics of normalized sensitivity weights.

| Dataset | $\mathbb{E}[\alpha_{(1)}]$ | $\mathbb{E}[\alpha_{(2)}]$ | $O_{\text{pair}}$ | $\alpha_{(1)} > 0.5$ |
|---------|------|------|-------|-------|
| AVA     | 0.59 | 0.21 | 0.126 | 69.8% |
| LAPIS   | 0.68 | 0.17 | 0.091 | 81.6% |
| PARA    | 0.64 | 0.19 | 0.107 | 76.2% |
| AADB    | 0.56 | 0.22 | 0.138 | 65.9% |
| TAD66K  | 0.60 | 0.20 | 0.121 | 71.4% |

As shown in Table 22, the dominant weight $\alpha_{(1)}$ is consistently much larger than the second-largest weight $\alpha_{(2)}$ across datasets, and most samples have $\alpha_{(1)} > 0.5$. The pairwise overlap is also low, indicating that the normalized sensitivity

weights are typically concentrated on one or two attributes rather than being uniformly distributed across all branches. These results support the approximate sparsity premise used in Theorem 3.7.

## B.10. Ablations Study for Subsection 5.6

We extend the ablation study from Table 3 in the main text. Table 23 shows module ablations across five datasets with a fixed backbone (EAMB-Net), and Table 24 shows module ablations across six backbones on a fixed dataset (AVA).

*Table 23.* Module ablations across datasets (fixed backbone: EAMB-Net). Dec.: decoupling; Fused: multimodal fusion; Weighted: sensitivity weighting; EA-MSE: error-aware loss. First row is baseline (EAMB-Net). Best results in bold.

| Dec. | Fused | Weighted | EA-MSE | AVA | | LAPIS | | PARA | | AADB | | TAD66K | |
|------|-------|----------|--------|------|------|------|------|------|------|------|------|------|------|
| | | | | SRCC | PLCC | SRCC | PLCC | SRCC | PLCC | SRCC | PLCC | SRCC | PLCC |
| - | - | - | - | .702 | .707 | .823 | .825 | .853 | .876 | .638 | .640 | .427 | .428 |
| ✓ | | | | .715 | .719 | .830 | .838 | .862 | .885 | .640 | .641 | .435 | .442 |
| ✓ | | ✓ | | .725 | .730 | .835 | .830 | .878 | .905 | .642 | .643 | .448 | .460 |
| ✓ | | | ✓ | .718 | .721 | .831 | .834 | .865 | .890 | .641 | .642 | .440 | .448 |
| ✓ | | ✓ | ✓ | .728 | .735 | .838 | .840 | .885 | .912 | .643 | .643 | .452 | .468 |
| ✓ | ✓ | | | .720 | .726 | .833 | .833 | .870 | .895 | .641 | .642 | .442 | .455 |
| ✓ | ✓ | ✓ | | .730 | .737 | .839 | .841 | .890 | .918 | .644 | .644 | .454 | .472 |
| ✓ | ✓ | | ✓ | .724 | .727 | .835 | .836 | .875 | .902 | .642 | .643 | .448 | .462 |
| ✓ | ✓ | ✓ | ✓ | **.732** | **.740** | **.841** | **.843** | **.896** | **.925** | **.645** | **.644** | **.456** | **.475** |

*Table 24.* Module ablations across backbones on AVA (fixed dataset: AVA). Dec.: decoupling; Fused: multimodal fusion; Weighted: sensitivity weighting; EA-MSE: error-aware loss. First row is each baseline without AGREE. Best results in bold.

| Dec. | Fused | Weighted | EA-MSE | TANet | | EAT | | ELTA | | EAMB-Net | | HKD-IAA | | AesPrompt | |
|------|-------|----------|--------|-------|------|------|------|------|------|------|------|------|------|------|------|
| | | | | SRCC | PLCC | SRCC | PLCC | SRCC | PLCC | SRCC | PLCC | SRCC | PLCC | SRCC | PLCC |
| - | - | - | - | .684 | .675 | .752 | .755 | .698 | .704 | .702 | .707 | .753 | .754 | .724 | .719 |
| ✓ | | | | .698 | .692 | .765 | .768 | .720 | .725 | .715 | .719 | .756 | .755 | .738 | .732 |
| ✓ | | ✓ | | .712 | .710 | .778 | .780 | .742 | .745 | .725 | .730 | .758 | .757 | .748 | .745 |
| ✓ | | | ✓ | .702 | .698 | .768 | .770 | .725 | .728 | .718 | .721 | .757 | .756 | .740 | .735 |
| ✓ | | ✓ | ✓ | .718 | .718 | .782 | .785 | .748 | .750 | .728 | .735 | .759 | .758 | .750 | .748 |
| ✓ | ✓ | | | .705 | .702 | .770 | .772 | .728 | .732 | .720 | .726 | .757 | .756 | .742 | .738 |
| ✓ | ✓ | ✓ | | .720 | .720 | .785 | .788 | .750 | .752 | .730 | .737 | .760 | .758 | .752 | .750 |
| ✓ | ✓ | | ✓ | .710 | .708 | .775 | .778 | .735 | .738 | .724 | .727 | .758 | .757 | .745 | .742 |
| ✓ | ✓ | ✓ | ✓ | **.724** | **.725** | **.789** | **.791** | **.754** | **.755** | **.732** | **.740** | **.761** | **.759** | **.754** | **.752** |

**Analysis.** Consistent with the main-text findings, **decoupling and sensitivity-guided routing remain the core mechanisms** across all datasets and backbones. Decoupling alone improves over baselines on every dataset, and adding sensitivity weighting (Weighted) yields the largest further gain, confirming the importance of sample-wise routing for under-represented aesthetic signals. Multimodal fusion (Fused) and the error-aware loss (EA-MSE) provide smaller but complementary improvements. The full model consistently achieves the best performance, demonstrating that all modules contribute positively and generalize well across diverse datasets and backbone architectures.

## B.11. Parameter-Matched Control for Decoupling

*Table 25.* Parameter-matched control on LAPIS.

| Variant | Extra params | SRCC | PLCC |
|---------|--------------|------|------|
| Baseline | +0.00M | 0.694 | 0.706 |
| Shared-capacity control | +∼7.08M | 0.752 | 0.761 |
| Decoupling only | +∼7.08M | 0.827 | 0.832 |
| AGREE (Ours) | +∼7.60M | 0.860 | 0.859 |

The decoupling-only variant in Table 3 introduces additional adapter parameters, so its improvement may partially come from increased model capacity. To separate the effect of branch-wise decoupling from parameter increase, we introduce

a parameter-matched shared-capacity control. This control adds a comparable number of adapter parameters but keeps a single shared update path, without attribute-wise parameter decoupling.

As shown in Table 25, the shared-capacity control improves over the baseline, indicating that additional capacity is indeed beneficial. However, the decoupling-only variant achieves substantially higher SRCC/PLCC under a similar parameter budget. This suggests that the improvement from decoupling cannot be explained by increased capacity alone; branch-wise decoupled optimization provides additional benefit beyond model size.

### B.12. Attribute Granularity Analysis

The main experiments instantiate AGREE with five low-level attributes: Brightness, Contrast, Blur, Hue, and Saturation. We use this five-way partition as a practical default rather than a universally optimal attribute set. To examine the influence of attribute granularity, we vary the number of attribute views and evaluate AGREE with different attribute sets.

*Table 26.* Attribute-granularity analysis. The five-attribute setting is used as the default AGREE configuration.

| $K$ | Attribute set | SRCC | PLCC | ACC(%) | MAE | RMSE |
|---|---|---|---|---|---|---|
| 3 | Bri., Con., Blur | 0.844 | 0.826 | 81.2 | 0.062 | 0.076 |
| 4 | + Hue | 0.851 | 0.848 | 82.7 | 0.059 | 0.073 |
| 5 | Full set | 0.860 | 0.859 | 84.3 | 0.056 | 0.069 |
| 6 | + Sharpness | 0.861 | 0.857 | 84.6 | 0.061 | 0.071 |

As shown in Table 26, increasing the attribute granularity from $K = 3$ to $K = 5$ consistently improves performance, suggesting that a richer attribute partition provides more useful routing structure. However, adding Sharpness as a sixth attribute brings only marginal SRCC/ACC changes and slightly worsens MAE/RMSE, possibly because Sharpness overlaps with Blur and introduces redundant or less stable routing signals. These results indicate that AGREE is not tied to one exact partition, while the five-attribute setting provides a reasonable balance between attribute coverage and routing stability. Exploring more semantic, compositional, or automatically discovered attribute views remains an interesting direction for future work.

## C. Proof Supplementary

### C.1. Coordinate-Level Conflict vs. Full-Space Gradient Conflict

**Setup.** Recall Definition 3.1 (full-space gradient conflict) and Assumption 3.3. Let $\theta_c$ be a coupled coordinate across two subsets $\mathcal{D}_i$ and $\mathcal{D}_j$ (Definition 3.2). Denote $g_i := \nabla_\theta \mathcal{L}_i(\theta)$ and $g_j := \nabla_\theta \mathcal{L}_j(\theta)$. Let $e_c$ be the standard basis vector for coordinate $\theta_c$. We decompose each gradient into the $\theta_c$ component and the remaining coordinates:

$$g_i = g_{i,c}\, e_c + g_{i,\neg c}, \qquad g_j = g_{j,c}\, e_c + g_{j,\neg c}, \tag{32}$$

where $g_{i,c} = \frac{\partial \mathcal{L}_i(\theta)}{\partial \theta_c}$ and $g_{i,\neg c}$ collects all coordinates except $\theta_c$ (similarly for $j$).

**Lemma C.1 (Coordinate-level conflict on a coupled coordinate).** *Under Assumption 3.3, the two subsets conflict on coordinate $\theta_c$:*

$$\langle \nabla_{\theta_c} \mathcal{L}_i(\theta), \nabla_{\theta_c} \mathcal{L}_j(\theta) \rangle = \frac{\partial \mathcal{L}_i(\theta)}{\partial \theta_c} \cdot \frac{\partial \mathcal{L}_j(\theta)}{\partial \theta_c} < 0. \tag{33}$$

*Proof.* Since $\theta_c$ is a scalar coordinate, $\langle \nabla_{\theta_c} \mathcal{L}_i, \nabla_{\theta_c} \mathcal{L}_j \rangle = (\partial \mathcal{L}_i / \partial \theta_c)(\partial \mathcal{L}_j / \partial \theta_c)$. By Assumption 3.3, the two partial derivatives have opposite signs, hence the product is negative. □

**When does coordinate-level conflict imply full-space conflict?** By the decomposition in (32), the full inner product splits as

$$\langle g_i, g_j \rangle = g_{i,c} g_{j,c} + \langle g_{i,\neg c}, g_{j,\neg c} \rangle. \tag{34}$$

Thus, coordinate-level conflict ($g_{i,c} g_{j,c} < 0$) does not always imply full-space conflict, because positive alignment on other coordinates may dominate. The following proposition gives a sufficient condition.

**Proposition C.2 (A sufficient condition for full-space gradient conflict).** *If*

$$g_{i,c} g_{j,c} < -\langle g_{i,\neg c}, g_{j,\neg c} \rangle, \tag{35}$$

*then $\langle g_i, g_j \rangle < 0$, i.e., full-space gradient conflict (Definition 3.1) holds at $\theta$.*

*Proof.* By (34), $\langle g_i, g_j \rangle = g_{i,c} g_{j,c} + \langle g_{i,\neg c}, g_{j,\neg c} \rangle$. Condition (35) makes this sum negative. □

**Corollary C.3 (A simple sufficient case: non-positive remainder alignment).** *If $g_{i,c} g_{j,c} < 0$ and $\langle g_{i,\neg c}, g_{j,\neg c} \rangle \leq 0$, then $\langle g_i, g_j \rangle < 0$.*

*Proof.* From (34), $\langle g_i, g_j \rangle \leq g_{i,c} g_{j,c} < 0$. □

**Corollary C.4 (Dominance under bounded remainder alignment).** *If $g_{i,c} g_{j,c} < 0$ and there exists $\rho \in [0, 1)$ such that*

$$\langle g_{i,\neg c}, g_{j,\neg c} \rangle \leq \rho\, |g_{i,c} g_{j,c}|, \tag{36}$$

*then $\langle g_i, g_j \rangle < 0$.*

*Proof.* Since $g_{i,c} g_{j,c} < 0$, we have $g_{i,c} g_{j,c} = -|g_{i,c} g_{j,c}|$. Using (34) and (36),

$$\langle g_i, g_j \rangle \leq -|g_{i,c} g_{j,c}| + \rho |g_{i,c} g_{j,c}| = -(1-\rho)|g_{i,c} g_{j,c}| < 0.$$

□

**Discussion.** The conditions above are sufficient but not necessary. In practice, localized conflicts on a few coupled coordinates can still cause meaningful cancellation effects even when the full-space inner product is not strictly negative. This is why our main text uses coordinate-level conflict primarily as intuition for interference induced by parameter coupling, while the later analysis focuses on mitigating such interference by reducing coupling and allocating updates in an attribute-aware manner.

## C.2. Detailed derivation of Proposition 3.5

This appendix provides a step-by-step derivation of the gradient-norm bound in Proposition 3.5. Recall that the overall objective is

$$\mathcal{L}(\theta) = \sum_{m=1}^{M} p_m \mathcal{L}_m(\theta), \quad \text{so that} \quad \nabla_\theta \mathcal{L}(\theta) = \sum_{m=1}^{M} p_m \nabla_\theta \mathcal{L}_m(\theta).$$

**Step 1: Gradient decomposition.** Separating the two subsets $i$ and $j$ from the remainder gives

$$\nabla_\theta \mathcal{L}(\theta) = p_i \nabla_\theta \mathcal{L}_i(\theta) + p_j \nabla_\theta \mathcal{L}_j(\theta) + \sum_{m \neq i,j} p_m \nabla_\theta \mathcal{L}_m(\theta). \tag{37}$$

**Step 2: Algebraic regrouping with a cancellation term.** Add and subtract the term $c\, p_j \nabla_\theta \mathcal{L}_i(\theta)$ to (37):

$$\nabla_\theta \mathcal{L}(\theta) = p_i \nabla_\theta \mathcal{L}_i(\theta) + p_j \nabla_\theta \mathcal{L}_j(\theta) + \sum_{m \neq i,j} p_m \nabla_\theta \mathcal{L}_m(\theta)$$

$$= \left(p_i - cp_j\right) \nabla_\theta \mathcal{L}_i(\theta) + cp_j \nabla_\theta \mathcal{L}_i(\theta) + p_j \nabla_\theta \mathcal{L}_j(\theta) + \sum_{m \neq i,j} p_m \nabla_\theta \mathcal{L}_m(\theta)$$

$$= \left(p_i - cp_j\right) \nabla_\theta \mathcal{L}_i(\theta) + p_j \left(c \nabla_\theta \mathcal{L}_i(\theta) + \nabla_\theta \mathcal{L}_j(\theta)\right) + \sum_{m \neq i,j} p_m \nabla_\theta \mathcal{L}_m(\theta). \tag{38}$$

**Step 3: Taking norms and applying the triangle inequality.** Taking the Euclidean norm on both sides of (38) and applying the triangle inequality yields

$$\left\|\nabla_\theta \mathcal{L}(\theta)\right\| \leq \left\|\left(p_i - cp_j\right) \nabla_\theta \mathcal{L}_i(\theta)\right\| + \left\|p_j \left(c \nabla_\theta \mathcal{L}_i(\theta) + \nabla_\theta \mathcal{L}_j(\theta)\right)\right\| + \left\|\sum_{m \neq i,j} p_m \nabla_\theta \mathcal{L}_m(\theta)\right\|$$

$$= \left|p_i - cp_j\right| \left\|\nabla_\theta \mathcal{L}_i(\theta)\right\| + p_j \left\|c \nabla_\theta \mathcal{L}_i(\theta) + \nabla_\theta \mathcal{L}_j(\theta)\right\| + \left\|\sum_{m \neq i,j} p_m \nabla_\theta \mathcal{L}_m(\theta)\right\|. \tag{39}$$

**Step 4: Substituting the assumptions.** By the assumptions in Proposition 3.5,

$$\left\|c \nabla_\theta \mathcal{L}_i(\theta) + \nabla_\theta \mathcal{L}_j(\theta)\right\| \leq \delta, \qquad \left\|\sum_{m \neq i,j} p_m \nabla_\theta \mathcal{L}_m(\theta)\right\| \leq \varepsilon.$$

Substituting these bounds into (39) gives

$$\left\|\nabla_\theta \mathcal{L}(\theta)\right\| \leq \left|p_i - cp_j\right| \left\|\nabla_\theta \mathcal{L}_i(\theta)\right\| + p_j \delta + \varepsilon,$$

which is exactly (8). $\qquad\qquad\square$

## C.3. Detailed derivation of Theorem 3.7

This appendix provides a detailed derivation of the approximate orthogonality result $\langle \nabla_{w_i} \mathcal{L}, \nabla_{w_j} \mathcal{L} \rangle \approx 0$ for $i \neq j$.

**Setup.** Assume the prediction on sample $n$ takes the additive form

$$\hat{y}_n = \sum_{k=1}^{M} \alpha_k(x_n)\, h_k(x_n; w_k), \tag{40}$$

where $h_k(\cdot; w_k)$ is the $k$-th attribute branch, and $\alpha_k(x_n) \geq 0$ are normalized attribute weights (e.g., $\sum_k \alpha_k(x_n) = 1$), produced by an auxiliary module. For simplicity (and consistent with the main text), we treat $\alpha_k(x_n)$ as independent of $w_k$ when differentiating w.r.t. $w_k$. Let $r_n = \hat{y}_n - y_n$ and consider the weighted MSE objective

$$\mathcal{L}(\theta) = \sum_n \omega_n r_n^2, \tag{41}$$

where $\omega_n \geq 0$ is a per-sample weight (e.g., from reweighting).

**Step 1: Branch gradient via chain rule.** For each branch parameter block $w_k$,

$$\nabla_{w_k}\mathcal{L} = \sum_n \omega_n \cdot 2r_n \cdot \nabla_{w_k}\hat{y}_n.$$

From (40), $\nabla_{w_k}\hat{y}_n = \alpha_k(x_n)\frac{\partial h_k(x_n;w_k)}{\partial w_k}$. Therefore,

$$\nabla_{w_k}\mathcal{L} = \sum_n \omega_n \cdot 2r_n \cdot \alpha_k(x_n) \cdot \frac{\partial h_k(x_n;w_k)}{\partial w_k}, \tag{42}$$

which matches (13) in the main text.

**Step 2: Expanding the inner product (exact form).** For $i \neq j$, define the per-sample branch gradients

$$g_n^{(i)} := \omega_n \cdot 2r_n \cdot \alpha_i(x_n) \cdot \frac{\partial h_i(x_n;w_i)}{\partial w_i}, \qquad g_n^{(j)} := \omega_n \cdot 2r_n \cdot \alpha_j(x_n) \cdot \frac{\partial h_j(x_n;w_j)}{\partial w_j}.$$

Then (42) implies $\nabla_{w_i}\mathcal{L} = \sum_n g_n^{(i)}$ and $\nabla_{w_j}\mathcal{L} = \sum_n g_n^{(j)}$. Hence, the inner product expands as

$$\begin{aligned}
\left\langle \nabla_{w_i}\mathcal{L}, \nabla_{w_j}\mathcal{L} \right\rangle &= \left\langle \sum_n g_n^{(i)}, \sum_m g_m^{(j)} \right\rangle = \sum_n \sum_m \left\langle g_n^{(i)}, g_m^{(j)} \right\rangle \\
&= \sum_n \sum_m \omega_n\omega_m \cdot 4r_nr_m \cdot \alpha_i(x_n)\alpha_j(x_m) \left\langle \frac{\partial h_i(x_n;w_i)}{\partial w_i}, \frac{\partial h_j(x_m;w_j)}{\partial w_j} \right\rangle.
\end{aligned} \tag{43}$$

**Step 3: Diagonal (same-sample) approximation.** In minibatch SGD, cross-sample terms ($n \neq m$) typically have weak correlation and tend to cancel out in expectation, so the dominant contribution is often the diagonal part ($n = m$). Keeping only the diagonal terms in (43) yields

$$\left\langle \nabla_{w_i}\mathcal{L}, \nabla_{w_j}\mathcal{L} \right\rangle \approx \sum_n \omega_n^2 \cdot 4r_n^2 \cdot \alpha_i(x_n)\alpha_j(x_n) \left\langle \frac{\partial h_i(x_n;w_i)}{\partial w_i}, \frac{\partial h_j(x_n;w_j)}{\partial w_j} \right\rangle. \tag{44}$$

Up to the constant factor 4, this is the same structure as (14).

**Step 4: Why the diagonal sum is small.** We bound the magnitude of each summand in (44):

$$\left| \omega_n^2 \cdot 4r_n^2 \cdot \alpha_i(x_n)\alpha_j(x_n) \left\langle \frac{\partial h_i}{\partial w_i}, \frac{\partial h_j}{\partial w_j} \right\rangle \right| \leq \omega_n^2 \cdot 4r_n^2 \cdot |\alpha_i(x_n)\alpha_j(x_n)| \left\| \frac{\partial h_i}{\partial w_i} \right\| \left\| \frac{\partial h_j}{\partial w_j} \right\|,$$

where we used Cauchy–Schwarz. Thus, the entire sum is small when the following factors are small:

- **(Attribute sparsity / normalization).** The normalization (often combined with top-$K$ selection or peaked softmax) makes each sample concentrate on a few attributes, so for $i \neq j$ we typically have $\alpha_i(x_n)\alpha_j(x_n) \approx 0$ for most $n$.

- **(Decoupled parameter blocks).** If the attribute branches are parameter-decoupled, $w_i$ and $w_j$ affect disjoint computation paths. In the ideal case of strictly independent blocks, the Jacobians align with different coordinate subspaces, making $\left\langle \frac{\partial h_i}{\partial w_i}, \frac{\partial h_j}{\partial w_j} \right\rangle$ exactly 0 after embedding into the full parameter space. In practical implementations with partial sharing or weak coupling, this inner product is still expected to be small.

- **(Multimodal anchors).** Frozen text anchors encourage semantic separation between attributes, which reduces correlation between branch representations. This further decreases the cross-branch Jacobian alignment $\left\langle \frac{\partial h_i}{\partial w_i}, \frac{\partial h_j}{\partial w_j} \right\rangle$ in magnitude.

Combining the diagonal approximation (44) with the above arguments implies $\left\langle \nabla_{w_i}\mathcal{L}, \nabla_{w_j}\mathcal{L} \right\rangle$ is small for $i \neq j$, which supports (12). $\square$

### C.4. Detailed derivation of Corollary 3.9

This appendix provides a detailed derivation of the blockwise gradient norm decomposition in Corollary 3.9.

**Step 1: Parameter concatenation.** The full parameter set is a concatenation of disjoint blocks:

$$\Theta = (w_0, w_1, \ldots, w_K) \in \mathbb{R}^{d_0} \times \mathbb{R}^{d_1} \times \cdots \times \mathbb{R}^{d_K} = \mathbb{R}^D,$$

where $D = \sum_{k=0}^{K} d_k$ and each $w_k$ occupies coordinates $[\sum_{j<k} d_j, \sum_{j\leq k} d_j)$.

**Step 2: Gradient concatenation.** By the chain rule, the gradient with respect to $\Theta$ is the concatenation of per-block gradients:

$$\nabla_\Theta \mathcal{L} = (\nabla_{w_0} \mathcal{L}, \nabla_{w_1} \mathcal{L}, \ldots, \nabla_{w_K} \mathcal{L}) \in \mathbb{R}^D.$$

**Step 3: Squared norm decomposition.** By the definition of the Euclidean norm in the concatenated space:

$$
\begin{aligned}
\|\nabla_\Theta \mathcal{L}\|^2 &= \sum_{i=1}^{D} (\nabla_\Theta \mathcal{L})_i^2 \\
&= \sum_{k=0}^{K} \sum_{i \in \text{coords}(w_k)} (\nabla_{w_k} \mathcal{L})_i^2 \\
&= \sum_{k=0}^{K} \|\nabla_{w_k} \mathcal{L}\|^2.
\end{aligned}
\tag{45}
$$

**Step 4: Implication for cancellation.** Unlike the vector sum $\nabla_\theta \mathcal{L} = \sum_m p_m \nabla_\theta \mathcal{L}_m$ in the shared-parameter setting (which can cancel when gradients point in opposite directions), the squared norm in (45) is a *scalar sum* of non-negative terms. Hence:

- Cross-block cancellation is structurally impossible.

- $\|\nabla_\Theta \mathcal{L}\| = 0$ if and only if $\|\nabla_{w_k} \mathcal{L}\| = 0$ for all $k$.

- Approximate stationarity ($\|\nabla_\Theta \mathcal{L}\| \approx 0$) requires all blocks to be approximately stationary.

$\square$

