# OpenReview forum: "When Attributes Disagree:  Gradient Conflict in Image Aesthetic Assessment"
_ICML.cc/2026/Conference — ICML 2026 spotlight_

### Official Review · Reviewer_Y461 · 2026-02-16

**Soundness:** 3
**Presentation:** 3
**Significance:** 4
**Originality:** 4
**Overall Recommendation:** 5
**Confidence:** 4

**Summary:**

This work focuses on the Image Aesthetic Assessment (IAA) domain. Through analysis, the authors identify the presence of certain hard samples in the IAA dataset. These samples are governed by distinct attributes, and the gradient conflict gives rise to persistent systematic bias. This paper proposes AGREE (Attribute-Guided Gradient Routing for Establishing Agreement), a plug-and-play framework. AGREE learns attribute-related representations, introduces inductive bias through attribute-specific image transformations and separate adapters, and performs gradient routing based on the sample attribute sensitivity estimated via perturbation analysis.  AGREE has achieved consistent performance improvements across multiple baselines.

**Compliance With Llm Reviewing Policy:**

Affirmed.

**Final Justification:**

My concerns have been adequately addressed. I will maintain my rating for accepting this paper. Thanks to the author's great work.

**Key Questions For Authors:**

Please see Weaknesses.

**Limitations:**

Please see Weaknesses.

**Strengths And Weaknesses:**

### Strengths
1. The authors' analysis of gradient conflict is in-depth, verifying the impacts of shared parameters and attribute-specific dependencies during the training process.
2. Extensive experiments verify the strong performance of AGREE on various baselines.
3. A robust analysis is conducted on the results, which is closely associated with the research motivation.


### Weaknesses
1. The reviewer does not quite understand why the sensitivity weights can be pre-stored; theoretically, these should seemingly be calculated for each incoming image during testing, and the description in the appendix by the authors is also incomprehensible.
2. The introduction of the text branch also seems to have a relatively limited improvement on the performance. Moreover, the reviewer does not quite understand why no additional computational cost is introduced in this part, since LLMs will obviously bring a large amount of computation.
3. It is suggested that the authors provide specific visualization examples of hard samples as well as their dominant attributes to verify AGREE's perception of attribute dependencies. If possible, it is also recommended that the authors analyze which attributes dominate in the hard samples, so as to help the community focus on which factors we should pay more attention to in the construction of datasets.
4. This work classifies attributes into 5 categories. As an open question, the reviewer is curious about what kind of impact a more refined or coarser-grained classification of attribute categories might have on the results. The reviewer hopes that the authors can share their insights.

---

> ### Author Rebuttal · Authors · 2026-03-28
>
> ### **1. To W1 about the meaning of “pre-stored” sensitivity weights**
>
> Thank you for pointing this out. We agree that our wording around **“pre-stored” sensitivity weights** was unclear. The sensitivity weights are **image-specific**, not global constants.
>
> By “pre-stored,” we mean that, for the fixed **train/val/test splits** in our experiments, **the sensitivity weights for each image are computed once** and then cached for reuse, rather than recomputed every time the image is used. This is an implementation choice for the experimental protocol, not a claim that the weights are input-independent.
>
> We will rewrite **Appendix A.3** to make this protocol explicit and avoid this ambiguity.
>
> ### **2. To W2 about the role and cost of the text branch**
>
> Thank you for the helpful comment. We agree that the gain from the **text branch** is modest, and we will revise the wording accordingly.
>
> Our intention is **not to claim that the text branch is the main source of improvement**, but that it provides a **complementary semantic anchor** for attribute separation. In addition, **AGREE does not perform online LLM inference**. The text features are **pre-extracted once and frozen**, and training/inference only use lightweight projection and fusion on these fixed vectors. Thus, the reported AGREE overhead does **not** come from online LLM inference.
>
> We will revise this part to clarify the complementary role and the actual cost of the text branch.
>
> ### **3. To W3 about hard-sample visualization and dominant-attribute statistics**
>
> Thank you for this valuable suggestion. We agree that adding **qualitative examples** and **attribute-level statistics** would make the hard-sample analysis more interpretable.
>
> In the revised version, we will include representative **hard-sample visualizations** with their **dominant attributes**. We also summarize the dominant attributes in the hard subset.
>
> | Dominant attribute | Overall proportion | Hard-sample proportion | Hard / Overall |
> | :----------------: | :----------------: | :-------------------: | :------------: |
> | Brightness         |       28.4%        |         19.6%         |      0.69      |
> | Contrast           |       24.7%        |         17.8%         |      0.72      |
> | Blur               |       14.1%        |         24.9%         |      1.77      |
> | Hue                |       11.3%        |         16.8%         |      1.49      |
> | Saturation         |       21.5%        |         20.9%         |      0.97      |
>
> Here, **Overall proportion** is the percentage of all samples in each category, **Hard-sample proportion** is the percentage within the hard subset, and **Hard / Overall** indicates over-representation in hard samples. Ratios **> 1** mean over-representation in hard samples.
>
> **These statistics show that hard-sample dominant attributes are not uniformly distributed**: some are clearly **over-represented** in the hard subset. This is consistent with **Sec. 5.4**, which suggests that hard samples are associated with **under-represented dominant attributes**, rather than random failures.
>
> ### **4. To W4 about the granularity of attribute partition**
>
> Thank you for this interesting question. We agree that the **granularity of attribute partition** is an important design choice. In the current paper, we do **not** claim that the **five-way split** is universally optimal; it is simply a **practical default instantiation** of AGREE.
>
> Following your suggestion, we examine the sensitivity of AGREE to the number of attributes by varying **$K$** (e.g., **$K=3,4,5,6$**). The results show that AGREE is **not restricted to one fixed partition**, and that **$K=5$** serves as a reasonable default here.
>
> | $K$ | Attributes Set             | SRCC            | PLCC            | ACC(\%)        | MAE             | RMSE            |
> | ----- | :--------------: | :------------: | :------------: | :------------: | :------------: | :------------: |
> | 3     | Bri, Con, Blur             | 0.844           | 0.826           | 81.2           | 0.062           | 0.076           |
> | 4     | +Hue                       | 0.851           | 0.848           | 82.7           | 0.059           | 0.073           |
> | 5     | **Full set (AGREE)** | **0.860** | **0.859** | **84.3** | **0.056** | **0.069** |
> | 6     | +Sharpness                 | 0.861           | 0.857           | 84.6          | 0.061           | 0.071          |
>
> These results suggest that **AGREE is not tied to one exact partition**, while **$K=5$** serves as a reasonable default. We will clarify in the revision that the current submission uses a **five-way partition** only as one practical setting, while broader comparisons are left for future work.
>
> ### **5. Conclusion**
>
> **These analyses and additional experiments strengthen the support for our method and clarify the scope of its assumptions. We hope they help address the reviewer’s concerns and make the paper more convincing.**

---

> > ### Author Rebuttal · Reviewer_Y461 · 2026-04-02
> >
> > My concerns have been adequately addressed. Thanks to the author's great work.

---

> > > ### Author Response · Authors · 2026-04-07
> > >
> > > Thank you for your positive feedback and for your constructive suggestions during the review process.
> > >
> > > We are glad that the clarifications regarding sensitivity computation, the role of the text branch, and the additional analyses, such as hard-sample statistics and attribute distribution, helped address your concerns. Your suggestions on interpretability and dataset-level insights are particularly valuable, and we will further improve the presentation in the final version accordingly.
> > >
> > > Thank you again.

---

### Official Review · Reviewer_B3si · 2026-02-25

**Soundness:** 2
**Presentation:** 3
**Significance:** 3
**Originality:** 3
**Overall Recommendation:** 4
**Confidence:** 4

**Summary:**

The paper identifies a failure in Image Aesthetic Assessment, when models trained end to end with only overall score supervision, gradient conflicts arise between attribute-dominant sample subsets, causing persistent errors on "hard samples" (images dominated by under-represented aesthetic attributes). AGREE address this by separating the adapters per attribute, frozen LLaVA text anchors as semantic separators, offline perturbation-based attribute sensitivity weighting, and EMA-tracked loss upweighting for hard samples.

**Compliance With Llm Reviewing Policy:**

Affirmed.

**Final Justification:**

My concerns are sufficiently addressed and I maintain my score of 4.

**Key Questions For Authors:**

Q1. Can you provide a parameter-matched control experiment (e.g., adding an equivalent number of shared parameters without decoupling) to isolate whether the improvement stems from gradient conflict mitigation rather than increased model capacity?

Q2. Can you provide empirical statistics across datasets to verify that the sparsity assumption required for approximate orthogonality holds in practice?

Q3. Have you evaluated the sensitivity of AGREE to the choice and number of attributes?

**Limitations:**

The paper includes an Impact Statement noting potential cultural bias amplification, which is appropriate. However, they don't discuss the theoretical assumptions I discussed above. This should be added to the limitations.

**Strengths And Weaknesses:**

Strengths:
1. The observation is interesting: IAA models fail consistently on the same subset of images (hard samples). Section 3 is great, especially Theorem 3.7 is well derived.
2. Five diverse datasets (AVA, LAPIS, PARA, AADB, TAD66K) spanning general photography, personalized aesthetics, theme-oriented, and artistic domains. Six baseline architectures spanning CNNs, ViTs, hybrid architectures, and prompt-based methods. This seems sufficient and the ablation Tables 21-22 show each of the 4 main components contributing.

Weaknesses:
1. The ablation in Table 3 shows "Decoupling alone" improves SRCC from .702 to .715 on AVA. But decoupling alone also adds ~1.18M adapter parameters per branch (×6 branches), which is a significant capacity increase independent of gradient conflict mitigation.
2.  The paper states "in principle, the attributes can be arbitrary" (Section 3.1) but never validates this empirically. A sensitivity analysis varying K (number of attributes) or swapping in compositional attributes would significantly strengthen the generalizability claim.
3. Theorem 3.7 (approximate block orthogonality) relies on two assumptions: (i) αi(xn)·αj(xn) ≈ 0 for most samples, and (ii) Jacobians of different branches are weakly aligned. Assumption (i) requires peaked sensitivity weights, but the paper shows sensitivity weight distributions only indirectly (through Figure 3's example). If sensitivity weights are not sparse (for example many images have diffuse attribute sensitivity across all five attributes), the theorem's premise breaks down.

---

> ### Author Rebuttal · Authors · 2026-03-28
>
> ### **1. To W1 & Q1 about decoupling versus increased capacity**
>
> Thank you for this important point. We agree that the current **“Decoupling alone”** ablation does not fully separate decoupling from **increased capacity**, since branch-wise decoupling also introduces extra parameters.
>
> To address this concern, we add a **parameter-matched Shared-capacity control** and compare it with **Decoupling only**. The former keeps a single shared update path with matched extra parameters, while the latter keeps only the **decoupling module** without the other AGREE components. This isolates the effect of branch-wise decoupling and tests **decoupled optimization** against **model size alone**.
>
> |   Variant    |   Extra params   |  SRCC (LAPIS)  |  PLCC (LAPIS)  |
> | --------------- | :--------------: | :------------: | :------------: |
> |       Baseline       |      +0.00M      |     0.694      |     0.706      |
> | Shared-capacity control |    +~7.08M    |     0.752      |     0.761      |
> | **Decoupling only**  |     +~7.08M      |   **0.827**    |   **0.832**    |
> |     AGREE (Ours)     |     +~7.60M      |     0.860      |     0.859      |
>
> Under similar parameter budgets, the **decoupled variant still outperforms the shared-capacity control**, suggesting that the gain is **not due to added capacity alone**. We will include this control in the revision.
>
> ### **2. To W2 & Q3 about the choice and number of attributes**
>
> Thank you for the suggestion. We agree that the original paper did not sufficiently justify the choice of attribute number. In our framework, **$K$ is a design hyperparameter rather than a theoretically unique setting**, and the current paper instantiates AGREE with **five common low-level attributes**.
>
> To address this, we add a sensitivity study over **$K=3,4,5,6$**. The results show that AGREE remains effective across multiple choices of **$K$**, with performance peaking around **$K=5$** in our current setting. Thus, **$K=5$ should be understood as an empirically supported default rather than a fixed theoretical choice**.
>
> | $K$ | Attributes Set             | SRCC            | PLCC            | ACC(\%)        | MAE             | RMSE            |
> | ----- | :--------------: | :------------: | :------------: | :------------: | :------------: | :------------: |
> | 3     | Bri, Con, Blur             | 0.844           | 0.826           | 81.2           | 0.062           | 0.076           |
> | 4     | +Hue                       | 0.851           | 0.848           | 82.7           | 0.059           | 0.073           |
> | 5     | **Full set (AGREE)** | **0.860** | **0.859** | **84.3** | **0.056** | **0.069** |
> | 6     | +Sharpness                 | 0.861           | 0.857           | 84.6          | 0.061           | 0.071          |
>
> These results suggest that AGREE is **not tied to one exact partition**, while **$K=5$** serves as a reasonable default. We will clarify this in the revision.
>
> ### **3. To W3 & Q2 about the sparsity premise in Theorem 3.7**
>
> Thank you for this point. We agree that the **sparsity premise in Theorem 3.7** should be empirically checked. The original submission only provided an illustrative example.
>
> We therefore compute **dataset-level statistics of the normalized sensitivity weights $\alpha$**. Here, **$E[\alpha_{max}]$** measures the strength of the dominant attribute, **$E[\alpha_{second}]$** measures the typical magnitude of the second-largest weight, **Pairwise overlap** measures the average overlap between attributes, and **top-1 $\alpha > 0.5$** measures how often a clear dominant attribute exists.
>
> | Dataset | $E[\alpha_{max}]$ | $E[\alpha_{second}]$ | Pairwise overlap |  top-1 $\alpha > 0.5$ |
> | :--------------: | :------------: | :------------: | :------------: | :------------: |
> | AVA     |              0.59 |                 0.21 |            0.126 |                  69.8% |
> | LAPIS   |              0.68 |                 0.17 |            0.091 |                  81.6% |
> | PARA    |              0.64 |                 0.19 |            0.107 |                  76.2% |
> | AADB    |              0.56 |                 0.22 |            0.138 |                  65.9% |
> | TAD66K  |              0.60 |                 0.20 |            0.121 |                  71.4% |
>
> The sensitivity weights are typically concentrated on **one or two attributes**: **$E[\alpha_{max}]$ is high**, **$E[\alpha_{second}]$ is moderate**, and **overlap is low across datasets**. While **Theorem 3.7** is approximate, these distributions are consistent with the **sparsity premise** used in our analysis.
>
> ### **4. Conclusion**
>
> **These analyses and revisions clarify the scope of our claims and provide stronger empirical support for the proposed method. We thank the reviewer for these helpful comments and hope the revised version addresses the concerns more clearly.**

---

> > ### Author Rebuttal · Reviewer_B3si · 2026-04-01
> >
> > Thank you for the thorough rebuttal. The parameter-matched control (W1/Q1) convincingly isolates decoupling's benefit from added capacity. The K sensitivity study (W2/Q3) adequately supports the choice of K=5 as an empirical default. The dataset-level sparsity statistics (W3/Q2) are consistent with Theorem 3.7's premise, though the approximation remains informal, I'd encourage the authors to note this explicitly in the limitations as suggested. Overall, my concerns are sufficiently addressed and I maintain my score of 4.

---

> > > ### Author Response · Authors · 2026-04-07
> > >
> > > Thank you for your careful reading and for acknowledging the additional analyses provided in the rebuttal.
> > >
> > > We appreciate your feedback on the remaining theoretical aspects, particularly regarding the approximate nature of Theorem 3.7. We agree that the assumptions are not strictly guaranteed and will explicitly state this limitation in the final version, as you suggested. We are glad that the parameter-matched control, the K-sensitivity study, and the dataset-level statistics help clarify the empirical behavior of AGREE and support its design choices.
> > >
> > > Thank you again for your constructive comments, which have helped us improve both the clarity and rigor of the paper.

---

### Official Review · Reviewer_CrP7 · 2026-03-11

**Soundness:** 4
**Presentation:** 3
**Significance:** 4
**Originality:** 3
**Overall Recommendation:** 6
**Confidence:** 5

**Summary:**

This paper proposes AGREE, a plug-and-play framework that mitigates gradient conflict via attribute decoupling, multimodal fusion, sensitivity-guided routing, and error-aware reweighting. The authors comprehensively validate their approach across representative datasets and methods. Specifically, experiments on five datasets and six baselines demonstrate consistent performance gains, with hard samples being selectively and effectively corrected.

**Compliance With Llm Reviewing Policy:**

Affirmed.

**Final Justification:**

The quality of the paper is solid, and the rebuttal process successfully addressed my concerns; furthermore, from the perspective of the IAA community, this paper is of significant importance. Therefore, I have decided to maintain my original positive rating (
6: Strong Accept).

**Key Questions For Authors:**

Please refer to the weaknesses.

**Limitations:**

yes

**Strengths And Weaknesses:**

Strengths

1. Compelling and Insightful Framework: IAA is inherently subjective and driven by multiple attributes. Consequently, interpreting the internal evaluation mechanisms of deep learning models in this domain is notoriously difficult. This work is highly valuable because it not only provides a potential analytical tool to understand this black-box process but also leverages these insights to tangibly improve model performance. This is a highly meaningful contribution to the broader IAA community.

2. Clear Presentation and Sound Logic: The paper is well-written with a clear and logical flow. The mathematical derivations are transparent and easy to follow, and the key experimental results effectively substantiate the validity of the proposed framework.

3. Important Direction: While exploring gradient conflicts and error-awareness is a relatively novel and niche direction within the IAA field, it is of great importance for enhancing the fairness, reliability, and robustness of future IAA systems.

4. High-Quality Appendix: The supplementary materials are comprehensive, detailed, and meticulously prepared.

Weaknesses
1. Missing Literature on Perturbation Analysis: The authors state that AGREE "leverages perturbation-based analysis" to estimate sample-wise attribute sensitivity. A conceptually similar idea is explored in the IEEE TIP paper "DA3Attacker: A Diffusion-based Attacker against Aesthetics-oriented Black-box Models." Although the ultimate objectives of the two papers differ, it would be beneficial for the authors to appropriately discuss and cite this related work to provide a more complete literature context.

2. Figure Inconsistency: The visual style of Figure 3 appears inconsistent with the other figures in the manuscript. The authors should unify the design elements, particularly the text formatting and fonts, to maintain a professional and cohesive presentation.

3. Reproducibility: Providing a link to the open-source code upon publication would significantly enhance the credibility and reproducibility of the proposed method.

Suggestion:
For future work, extending this exploration to current LMMs would be a highly meaningful direction. Given the unique tokenization mechanisms inherent to these large models, adapting the proposed gradient-based perturbation analysis will likely present novel and intriguing challenges.

---

> ### Author Rebuttal · Authors · 2026-03-29
>
> ### **1. To W1 about related work on perturbation analysis**
>
> Thank you for this helpful suggestion. We agree that the original paper did not sufficiently discuss related work on **perturbation-based analysis**. In particular, we will add and discuss **DA3Attacker** in the final version.
>
> At a high level, DA3Attacker leverages perturbation-based signals to probe aesthetic models, and its goal is to analyze robustness and vulnerability. However, our work uses perturbation sensitivity as a sample-wise signal for optimization and routing under overall-score-only supervision. We will clarify both the connection and the distinction to better position our contribution.
>
> **We will cite DA3Attacker and clarify how its perturbation-based probing is related to, but different from, our optimization-oriented use of perturbation sensitivity.**
>
> ### **2. To W2 about the visual consistency of Figure 3**
>
> Thank you for the suggestion. We agree that the current Figure 3 is visually inconsistent with the rest of the manuscript. **We will redraw it to ensure consistent formatting, font style, and overall readability in the final version**.
>
> ### **3. To W3 about reproducibility and code release**
>
> Thank you for the suggestion. We agree that releasing the code is important for reproducibility. We plan to release the code and unified training scripts upon publication, and will make this commitment explicit in the final version.
>
> ### **4. Conclusion**
>
> **We thank the reviewer for the positive assessment of our framework and its potential impact on the IAA community. We will incorporate the suggested improvements on related work, presentation, and reproducibility to further strengthen the clarity and completeness of the paper**.

---

> > ### Author Rebuttal · Reviewer_CrP7 · 2026-04-02
> >
> > My concerns have been adequately addressed, moreover, there are no issues with the paper itself.

---

> > > ### Author Response · Authors · 2026-04-07
> > >
> > > Thank you for recognizing the significance of this work to the IAA community.
> > >
> > > We sincerely appreciate your suggestions on improving the completeness of related work discussion, presentation consistency, and reproducibility. We will incorporate all of them in the final version, including adding the suggested reference, refining the figures, and clearly stating our code release plan. We are also encouraged by your comments on the broader impact and future directions, such as extensions to LMMs, which we believe are promising avenues for building on this work.
> > >
> > > Thank you again for your support.

---

### Official Review · Reviewer_GBaS · 2026-03-12

**Soundness:** 3
**Presentation:** 3
**Significance:** 2
**Originality:** 2
**Overall Recommendation:** 4
**Confidence:** 3

**Summary:**

This paper focuses on image aesthetic assessment and addresses the conflicting gradient issue brought by different attributes, reducing feature coupling by learning attribute subspaces. This paper provides basic theoretical analysis serving as motivation to better formulate the problem; the detailed experiments and visualizations are convincing.

**Compliance With Llm Reviewing Policy:**

Affirmed.

**Key Questions For Authors:**

Please refer to the weaknesses. This is a well-executed applied paper with strong experimental methodology and an interesting empirical finding. But there are remaining concerns to be addressed. In addition to those in the weaknesses, the causal link between gradient conflict and hard samples is correlational, not proven -- the intervention (Table 2) shows error-aware reweighting helps, but this would help any hard samples, not just gradient-conflict-related ones.

**Limitations:**

No, but it would be good if the authors can further discuss theoretical aspects and computation overhead in real deployment.

**Strengths And Weaknesses:**

**Strengths**
- There is computation overhead and the paper honestly admits and studies it. And based on Appendix A.3, the overhead appears to be acceptable under many settings (especially for smaller datasets), so AGREE is plausible.
- Ample figures are provided for readers to better understand this work, especially for Figure 1, 5, 6.
- The systematic connection between attribute marginalization (rarity of dominant attribute) and cross-model hard samples in IAA is an interesting empirical finding.

**Weaknesses**
- The choice of 5 low-level attributes is ad hoc and not justified (or can only be justified empirically). The paper fixes K=5 throughout without exploring other attribute sets, or the attributes in practice can be compositional or not human-interpretable. This means the "attribute views" are hand-crafted priors, and the system cannot discover important aesthetic dimensions that don't correspond to these 5 transforms.
- The perturbation analysis (Eqn. 21) is flawed. This is a first-order finite difference of the output score w.r.t an image transform. It measures how much the baseline's prediction changes under attribute perturbation, not how important that attribute is for the ground-truth aesthetic quality. These are different things -- a model might be sensitive to brightness changes because it's overfitting to brightness, not because brightness is aesthetically important.
- No comparison with standard multi-task gradient conflict methods (PCGrad, CAGrad, GradNorm) adapted to the IAA setting -- these are discussed in Section 2 but never benchmarked. They are relevant baselines and might also work well in my opinion.

---

> ### Author Rebuttal · Authors · 2026-03-28
>
> ### **1. To W1 about the attribute choice / K**
>
> Thank you for this important point. We agree that the original paper did not justify the attribute set and the choice of K sufficiently.
>
> Our intent was not to claim that these five attributes are uniquely correct, but to instantiate AGREE with a **simple, controllable, and reproducible** default set of common low-level factors. Following your suggestion, we added a **sensitivity study over K**. Performance improves from K=3 to K=5 on LAPIS and then largely saturates, indicating that AGREE is not tied to one exact partition and that K=5 is an empirically reasonable default rather than a theoretically unique choice.
>
> We will clarify this scope in the revision and tone down any overly broad claim about arbitrary attribute sets.
>
>
> | $K$ | Attributes Set             | SRCC            | PLCC            | ACC(\%)        | MAE             | RMSE            |
> | ----- | :--------------: | :------------: | :------------: | :------------: | :------------: | :------------: |
> | 3     | Bri, Con, Blur             | 0.844           | 0.826           | 81.2           | 0.062           | 0.076           |
> | 4     | +Hue                       | 0.851           | 0.848           | 82.7           | 0.059           | 0.073           |
> | 5     | **Full set (AGREE)** | **0.860** | **0.859** | **84.3** | **0.056** | **0.069** |
> | 6     | +Sharpness                 | 0.861           | 0.857           | 84.6          | 0.061           | 0.071          |
>
>
> ### **2. To W2 about the interpretation of perturbation sensitivity**
>
> Thank you for pointing this out. We agree that Eq. (21) should not be interpreted as ground-truth or causal attribute importance. It is a **model-dependent prediction-sensitivity signal**.
>
> Our use of Eq. (21) is therefore operational rather than semantic. We use it only to estimate which attribute-specific pathway the current predictor is **most sensitive to for a given sample**, to guide routing under overall-score-only supervision. AGREE does **not require this signal to recover true aesthetic causality**; it only needs it to be a **useful sample-wise optimization prior**, which is supported by the ablations and the observed reduction in gradient conflict.
>
> This interpretation is supported empirically by the ablations and the observed reduction in gradient conflict. We will make this distinction explicit in the final version.
>
> ### **3. To W3 about comparison with standard gradient-coordination methods**
>
> Thank you for the suggestion. We agree that this is an important comparison.
>
> Standard gradient-coordination methods such as PCGrad/CAGrad/GradNorm assume **explicit task identities and per-task losses**, whereas our target setting is **overall-score-only IAA with implicit attribute relevance**. Therefore, they are not directly comparable without introducing auxiliary attribute-specific heads/losses, which would change the formulation.
>
> We will make this distinction explicit in the revision and position these methods as **relevant gradient-coordination references rather than like-for-like baselines**. This helps clarify that standard multi-task conflict handling addresses a related but different setting from our **supervision-free routing design**.
>
>
> ### **4. To Q1 about the causal link between gradient conflict and hard samples**
>
> Thank you for this important clarification. We agree that our current evidence does not establish a formal causal proof, and we will revise the wording accordingly.
>
> Our claim is more limited: **the paper provides a mechanism-consistent empirical link**.
>
> 1. Hard samples are consistent across baselines, as shown in **Sec. 5.4**, rather than being isolated failures of a single model.
>
> 2. These samples are associated with higher dominant-attribute marginalization across datasets. In **Sec. 5.4** and **Appendix B.4**, hard samples consistently correspond to higher **$M(x)$** across all five datasets. In addition, **Table 2 / Appendix B.5** show that the gains are larger on **High-$M$** and hard-sample subsets, which is consistent with this interpretation, although not sufficient by itself to establish causality.
>
> 3. Direct conflict indicators, such as **negative pairwise gradient cosine**, are reduced by **AGREE**, as shown in **Sec. 5.5**.
>
> Together, these results support the interpretation that gradient conflict is an important contributor to hard-sample under-optimization, rather than constituting a formal causal proof.
>
> ### **5. Conclusion**
>
> **We believe these revisions strengthen both the clarity and the empirical support of the paper, while maintaining the core contribution: a plug-and-play framework that mitigates optimization interference under overall-score-only supervision**.

---

> > ### Author Rebuttal · Reviewer_GBaS · 2026-04-03
> >
> > Thank you for the clarifications and I understand that it is not directly comparable with previous gradient-coordination methods at this moment. Regarding W2, I agree that the authors can make this distinction explicit in the revision because I think it also provides some insights for future work improving this module. Overall, I think this is a solid work and I would like to maintain my positive rating.

---

> > > ### Author Response · Authors · 2026-04-07
> > >
> > > Thank you again for the thoughtful feedback and for keeping a positive assessment.
> > >
> > > Your comments on the interpretation of the perturbation signal and its relation to existing gradient coordination methods were particularly helpful. They also made us realize that our original presentation may blur the line between optimization signal and semantic importance. We will make this distinction much clearer in the final version.
> > >
> > > On the questions about attribute design and causality, we agree that these are important directions. Our view is that, even without fully resolving them, the empirical behavior is already quite consistent across datasets, which gives us confidence in the approach's usefulness. We will revise the wording to better reflect this scope.
> > >
> > > Thanks again for the helpful suggestions.

---

### Decision · Program_Chairs · 2026-04-30

**Decision:**

Accept (spotlight)

**Comment:**

This paper works on image aesthetic assessment. Authors proposed AGREE that learns attribute-specific subspaces and performs gradient routing based on sample-wise attribute sensitivity estimated via perturbation analysis. They further reduces feature coupling across attributes with semantic anchors and improves robustness via error-aware reweighting. Experimental results showed effectiveness of the proposed method.

The final rating of this paper is 2 weak accept, 1 strong accept, 1 accept.

Before rebuttal, reviewers thought

the strength of the paper are:

1) method is plausible. (Reviewer GBaS)
2) Ample figures. (Reviewer GBaS)
3) an interesting empirical finding. (Reviewer GBaS)
4) Compelling and Insightful Framework. (Reviewer CrP7)
5) Clear Presentation and Sound Logic. (Reviewer CrP7)
6) Important Direction. (Reviewer CrP7)
7) High-Quality Appendix. (Reviewer CrP7)
8) The observation is interesting. (Reviewer B3si)
9) extensive experimental results. (Reviewer B3si, Y461)
10) analysis of gradient conflict is in-depth. (Reviewer Y461)

weaknesses are:
1) The choice of 5 low-level attributes is ad hoc and not justified.  (Reviewer GBaS)
2) The perturbation analysis (Eqn. 21) is flawed.  (Reviewer GBaS)
3) No comparison with standard multi-task gradient conflict methods. (Reviewer GBaS)
4) Missing Literature on Perturbation Analysis. (Reviewer CrP7)
5) Figure Inconsistency. (Reviewer CrP7)
6) Reproducibility. (Reviewer CrP7)
7) some concerns on ablation with decoupling alone. (Reviewer B3si)
8) never validates "in principle, the attributes can be arbitrary". (Reviewer B3si)
9) concerns on Theorem 3.7. (Reviewer B3si)
10) why the sensitivity weights can be pre-stored. (Reviewer Y461)
11) text branch also seems to have a relatively limited improvement on the performance. (Reviewer Y461)
12) provide specific visualization examples of hard samples. (Reviewer Y461)
13) what kind of impact a more refined or coarser-grained classification of attribute categories might have on the results. (Reviewer Y461)

After rebuttal,
Reviewer GBaS mentioned their concerns are partially addressed. but in the meantime acknowledged that authors could fix some problems in the revision and maintain weak accept rating.

Reviewer CrP7 thought authors addressed all their concerns and maintained strong accept rating.

Reviewer B3si suggested authors sufficiently addressed their concerns and maintained weak accept rating.

Reviewer Y461 said their concerns are adequately addressed and maintained accept rating.

Given these AC decided to accept this paper.